# Transcriptional effects of copy number alterations in a large set of human cancers

Arkajyoti Bhattacharya [1,2], Rico D. Bense[1,2], Carlos G. Urzúa-Traslaviña [1], Elisabeth G.E. de Vries [1], Marcel A.T.M. van Vugt[1] & Rudolf S.N. Fehrmann [1]*

Copy number alterations (CNAs) can promote tumor progression by altering gene expression levels. Due to transcriptional adaptive mechanisms, however, CNAs do not always translate proportionally into altered expression levels. By reanalyzing >34,000 gene expression profiles, we reveal the degree of transcriptional adaptation to CNAs in a genome-wide fashion, which strongly associate with distinct biological processes. We then develop a platform-independent method—transcriptional adaptation to CNA profiling (TACNA profiling)—that extracts the transcriptional effects of CNAs from gene expression profiles without requiring paired CNA profiles. By applying TACNA profiling to >28,000 patient-derived tumor samples we define the landscape of transcriptional effects of CNAs. The utility of this landscape is demonstrated by the identification of four genes that are predicted to be involved in tumor immune evasion when transcriptionally affected by CNAs. In conclusion, we provide a novel tool to gain insight into how CNAs drive tumor behavior via altered expression levels.

[1] Department of Medical Oncology, University Medical Center Groningen, University of Groningen, Groningen, the Netherlands. [2]These authors contributed equally: Arkajyoti Bhattacharya, Rico D. Bense *email: r.s.n.fehrmann@umcg.nl

The genomic composition of a tumor results from defects in genome maintenance, leading to the genomic instability that characterizes many cancers[1]. Genomic instability can result in the accumulation of structural chromosome aberrancies, including copy-number alterations (CNAs). CNAs are important genomic events in the progressive molecular rewiring that takes place during tumor cell evolution[2].

CNAs can promote tumor progression via alteration of expression levels of genes located at the affected genomic regions[3–5]. Due to transcriptional adaptive mechanisms, however, changes in gene copy number at the genomic level do not always translate proportionally into altered gene expression levels (Fig. 1a)[6,7]. The degree of transcriptional adaptation to CNAs is currently determined with the genetical genomics approach[8],

**Fig. 1 Data acquisition and decomposition of gene expression profiles. a** CNAs can promote tumor progression via altering expression levels of genes located at the affected genomic regions. However, due to transcriptional adaptation mechanisms, changes in gene copy number at the genomic level do not always translate proportionally into altered mRNA expression levels. Unraveling the degree of transcriptional adaptation to CNAs for all genes will greatly contribute to our knowledge how CNAs drive tumor progression. **b** Number of gene expression profiles collected from GEO, TCGA, CCLE, and GDSC. **c** Identification of underlying regulatory factors of the mRNA transcriptome. We hypothesized that the observed gene expression in a gene expression profile is the result of (i) the effect of underlying regulatory factors (i.e., source signals) on expression levels of individual genes and (ii) the activity of these underlying regulatory factors in a complex biopsy (i.e., mixing matrix). ICA was used to capture the number and nature of these underling regulatory factors for all four datasets separately. This resulted in estimated sources, representing the effects of independent underlying regulatory factors on the expression levels of individual genes, and a mixing matrix reflecting the activity of each estimated source in each gene expression profile. ICA was run 25 times with random initialization, followed by consensus sources estimation using a credibility index ≥50%. **d** Examples of CNA-CESs harboring a pattern in which only genes mapping to a specific contiguous genomic region had a high absolute weight. The red line shows whether genomic regions were marked as having a significant number of genes with a high absolute weight by the detection algorithm (i.e., extreme-valued region indicator).

which combines paired genomic and gene expression profiles of tumor samples to explore associations between CNAs and gene expression levels. However, we previously demonstrated that this approach is difficult to use for detecting the effects of CNAs on gene expression levels in a gene expression profile[9]. This is because gene expression profiling is often performed on biopsies comprising tumor cells and non-tumor cells from the tumor microenvironment. A gene expression profile therefore measures the average expression of all cell types present in tumor biopsies. This means that the effects of CNAs on gene expression levels, besides being influenced by experimental and other non-genetic factors, are often overshadowed by the effects of non-tumor cells[10,11]. Therefore, to accurately determine the degree of transcriptional adaptation to CNAs using genetical genomics, large numbers of paired genomic and gene expression profiles of tumor samples are required. Unfortunately, few large-scale genetical genomic datasets are available, and many of the publicly accessible gene expression profiles from tumor samples do not have a paired genomic profile. These samples are therefore currently excluded from genetical genomics analyses. Despite the current efforts in genetical genomics, the degree of transcriptional adaptation to CNAs thus remains unclear for most genes.

Unraveling the degree of transcriptional adaptation to CNAs in a genome-wide fashion would greatly enhance our knowledge of how CNAs drive tumor progression. We therefore develop a new, platform-independent method: transcriptional adaptation to CNA profiling (TACNA profiling). TACNA profiling extracts the effects of CNAs on gene expression levels from a single gene expression profile—generated from a tumor biopsy—without the need for a paired genomic CNA profile. Using this method, we define the landscape of transcriptional effects of CNAs in >34,000 expression profiles. To demonstrate this landscape's utility, we determine which genes and biological pathways affected by CNAs might be associated with tumor-infiltrating immune cell composition. These insights could ultimately contribute to the development of new therapeutic strategies.

## Results

**Data acquisition**. We collected gene expression data of 34,494 samples from four public repositories: Gene Expression Omnibus (GEO, $n = 21,592$, generated with Affymetrix HG-U133 Plus 2.0), The Cancer Genome Atlas (TCGA, $n = 10,817$, generated with RNA-seq), Cancer Cell Line Encyclopedia (CCLE, $n = 1067$, generated with Affymetrix HG-U133 Plus 2.0), and Genomics of Drug Sensitivity in Cancer (GDSC, $n = 1018$, generated with Affymetrix HG-U219) (Fig. 1b). In total, 2085 expression profiles were generated from cell line samples, 28,200 from patient-derived tumor biopsies, and 4209 from patient-derived tissue biopsies of normal tissue. The patient-derived tumor gene expression profiles collected from GEO and TCGA represented 30 tumor types in total (Supplementary Data 1–4). A detailed

description of preprocessing, normalization and quality control is provided in the Supplementary Note 1.

**Independent sources capture transcriptional effects of CNAs.** In tumor gene expression profiling, the net measured expression level of an individual gene is determined by the combined effects of various transcriptional regulatory factors, including experimental, genetic (e.g., CNAs), and non-genetic factors. To identify the effects of these factors on gene expression levels, we first performed consensus-independent component analysis (consensus-ICA)[12] on each of the four datasets separately (Fig. 1c, see Supplementary Note 1). Consensus-ICA is a computational method to separate gene expression profiles into additive consensus estimated sources (CESs), so that each CES is statistically as independent from the other CESs as possible. We hypothesized that each CES describes the effect of a latent transcriptional regulatory factor on gene expression levels. In every CES, each gene has a weight that describes how strongly and in which direction its expression level is influenced by a latent transcriptional regulatory factor. Ultimately, consensus-ICA resulted in 855 CESs in the GEO dataset, 1383 CESs in the TCGA dataset, 467 CESs in the CCLE dataset, and 466 CESs in the GDSC dataset.

In all four datasets, many CESs showed a consistent pattern in which only genes mapping to a specific contiguous genomic region had a high absolute weight (Fig. 1d). We hypothesized that these CESs (referred to as CNA-CESs) capture the effects of CNAs on gene expression levels. Using a detection algorithm (see Supplementary Note 1), we identified 242/855 (28%) of CNA-CESs in the GEO dataset containing such a pattern, 447/1383 (32%) in the TCGA dataset, 186/467 (40%) in the CCLE dataset and 175/466 (38%) in the GDSC dataset. In these CNA-CESs, 97% of genes present in the GEO dataset, 97% in the TCGA dataset, 95% in the CCLE dataset, and 94% in the GDSC dataset were located in at least one genomic region identified by the detection algorithm as having a significant number of genes with a high absolute weight (Supplementary Data 5). These high percentages indicate that almost all genes are affected by CNAs in our datasets.

**TACNA profiles highly correlate with SNP-derived profiles.** To continue testing the above hypothesis, we developed a method called transcriptional adaptation to CNA profiling (TACNA profiling—see Supplementary Note 1). In TACNA profiling, the gene expression profile of an individual sample is reconstructed using only CNA-CESs. By including only these CNA-CESs, we automatically excluded other CESs that capture non-genetic factors that can affect gene expression levels from our pipeline for TACNA profiling (see Supplementary Fig. 1 and Supplementary Note 1). Weights of genes mapping to a contiguous genomic region in CNA-CESs marked by the detection algorithm were

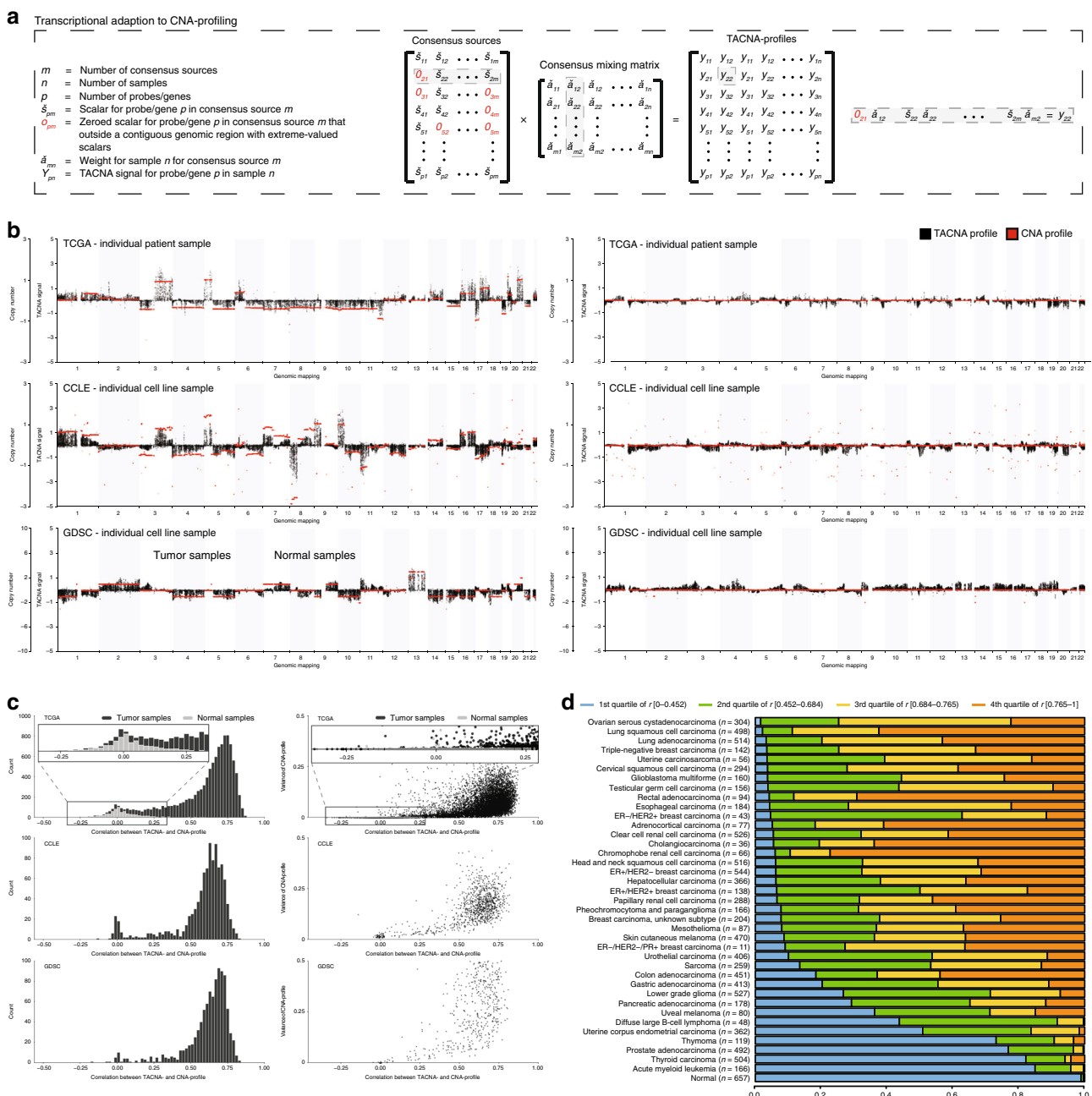

**Fig. 2 Transcriptional adaptation to CNA profiling (TACNA profiling). a** Transcriptional adaptation to CNA (TACNA) profiling. For each dataset, weights of genes mapping to a contiguous genomic region in CNA-CESs marked by the detection algorithm were retained in the CES matrix. Weights of genes that mapped outside marked genomic regions were instead set to zero. Next, TACNA profiles were calculated as the product between this transformed CES matrix and the consensus mixing matrix. **b** Examples of TACNA profiles illustrating that the resulting patterns clearly matched those in their paired, independently generated CNA profiles. **c** Left panel: distribution of Pearson correlations between TACNA profiles and paired CNA profiles for the TCGA, CCLE, and GDSC datasets. Right panel: variance observed in CNA profiles versus the Pearson correlation between CNA profiles and paired TACNA profiles for the TCGA, CCLE, and GDSC datasets. **d** Quartiles distribution plot of Pearson correlations between TACNA profiles and paired CNA profiles, per tumor type in the TCGA dataset. ER: estrogen receptor; HER2: human epidermal growth factor receptor 2; PR: progesterone receptor.

retained in the CNA-CES matrix. Weights of genes that mapped outside marked genomic regions in every CNA-CES were set to zero. Next, TACNA profiles were calculated as the product between this transformed CNA-CES matrix and the consensus mixing matrix (Fig. 2a).

We assumed that the resulting TACNA profile would show the effect of CNAs present in the genome on gene expression levels of that sample, and thus expected to see a clear correlation between paired TACNA and CNA profiles. We retrieved CNA profiles,

derived from SNP arrays, for a subset of samples in the TCGA dataset ($n = 10,620$), CCLE dataset ($n = 1011$) and the GDSC dataset ($n = 962$). Examples of TACNA profiles showed that the observed patterns clearly matched those in their paired, independently generated CNA profiles (Fig. 2b). Strong correlations were found between TACNA profiles and paired CNA profiles in all three datasets (Fig. 2c). Low correlations were found for a subset of samples in the TCGA dataset mainly representing normal tissue, which generally does not contain CNAs (see inset

Fig. 2c). As expected, we found lower correlations between paired TACNA and CNA profiles in the TCGA dataset for patient-derived samples belonging to the more point-mutation-driven tumor types (e.g., acute myeloid leukemia) than for samples from tumor types that are more copy-number-driven (e.g., ovarian cancer) (Fig. 2d)[13]. In a subset of patient-derived acute myeloid leukemia samples in the GEO dataset for which karyotype information was provided online ($n = 189$) we also observed clearly matching patterns between TACNA profiles and the expected karyotype in these samples (Supplementary Fig. 2).

We obtained gene expression profiles of an experimental model of aneuploidy introduced in HCT116 cells[14]. Differences on the mRNA level and TACNA level were calculated between the parental HCT116 cells ($n = 3$) and an isogenic model with chromosome 5 tetrasomy ($n = 3$) (Supplementary Fig. 3a and 3b). For genes located on chromosome 5, we observed larger signal to noise ratio in differences of TACNA expression levels compared with regular mRNA expression levels (Supplementary Fig. 3c). We did not observe a correlation between the log2 fold change in mRNA expression of individual genes and changes in TACNA expression levels ($r = 0.04$). We have previously shown that the effects of CNAs on gene expression levels are often overshadowed by other (non-genetic) effects that are not associated with specific single CNAs[9]. For example, Stingele et al. showed that aneuploidy invokes a general cellular mRNA response which is not the result of any specific CNAs but only the presence of aneuploidy[14]. In addition, tumor cells in vivo may develop transcriptional adaptive mechanisms to survive under selective pressure, which may be different from the adaptive mechanisms observed in vitro.

We conducted expression quantitative trait loci (eQTL) analyses using the mRNA expression profiles, the TACNA profiles and the CNA profiles (derived from SNP arrays) from the TCGA dataset ($n = 10,620$). In general, we observed a stronger correlation between TACNA profiles and CNA profiles compared with the correlation between mRNA expression profiles and CNA profiles on the gene level (Supplementary Fig. 4). This indicates that TACNA profiling increases statistical power to detect eQTLs, i.e., associations between CNAs and mRNA expression levels.

We observed that the borders of marked genomic regions in 82/242 (33%) CNA-CESs in the GEO dataset and 141/447 (32%) CNA-CESs in the TCGA dataset colocalized with a stringent set of common fragile sites on the human genome (Supplementary Data 6)[15]. Although the location of most of these common fragile sites have been mapped using low-resolution methods, these results are compatible with the notion of genomic instability underlying structural abnormalities such as CNAs.

Taken together, these results strongly suggest that TACNA profiles capture the effect of CNAs at the gene expression level.

**The inferred degree of TACNA is concordant across platforms**. Although strong correlations were found between paired TACNA and CNA profiles, not all genes mapping to a marked genomic region in a CNA-CES had equally high weights. We hypothesized that the absolute weight of an individual gene in a marked genomic region of a CNA-CES represents how much its expression level is affected by the underlying CNA: a lower absolute weight corresponds with a higher degree of transcriptional adaptation. A pair-wise comparison of the weights of genes mapping to marked genomic regions in CNA-CESs between the four datasets showed that many CNA-CESs strongly correlated with another CNA-CES in each of the three remaining datasets (Fig. 3a and Supplementary Data 7). This indicates that the specific patterns observed in many CNA-CESs are robust and independent of platform or dataset. For example, we found a high correlation

between two CNA-CESs in the GEO dataset and TCGA dataset in which the oncogene *KRAS* had a high weight (Fig. 3b, $r = 0.66$). Moreover, we performed TACNA profiling using 570 samples from the TCGA that were subjected to both RNA sequencing and microarray profiling (Affymetrix HG-U133A). Paired TACNA profiles generated with RNA-seq and microarray data were highly correlated (Supplementary Fig. 5, mode of distribution $r = 0.66$). In addition, TACNA profiling was robust to both cross-validation within one platform (Supplementary Figs. 6 and 7, mode of distribution $r$ range = 0.45–0.55) and cross-platform cross-validation (Supplementary Fig. 8, mode of distribution $r$ range = 0.50–0.61, see Supplementary Note 1 for details).

These results show that TACNA profiling can extract similar patterns from gene expression profiles generated with different platforms.

**The degree of TACNA is associated with biological processes**. Next, we determined whether the patterns observed in CNA-CESs were associated with biological processes rather than being mere mathematical artifacts. For each CNA-CES we transformed the weights of genes mapping to the marked genomic region to a metric ranging from zero to one, where zero corresponds to the highest absolute weight (i.e., low degree of transcriptional adaptation to CNAs) and one corresponds to the lowest absolute weight (i.e., high degree of transcriptional adaptation to CNAs). Subsequently, all genes mapping to marked genomic regions in the CNAs-CESs were pooled and ranked according to their metric. We then performed gene set enrichment analysis (GSEA) on this ranked gene list using 12 gene set databases from the Molecular Signatures Database (MSigDB)[16]. The analysis showed strong biological enrichment for all databases, with high concordance between GSEA results obtained with GEO and TCGA metrics ($r$ ranging between 0.86 and 0.97, Supplementary Data 8). Genes with a lower degree of transcriptional adaptation were highly enriched for gene sets involved in proliferation (Fig. 3c). Conversely, genes with a higher degree of transcriptional adaptation were highly enriched for immune-related gene sets. This indicates that the expression levels of genes involved in immune signaling are less affected on average when CNAs occur in their genomic region. The opposite effect was seen for genes involved in signaling pathways important for cell proliferation.

We observed that multiple genomic regions were enriched for genes having a higher or lower degree of transcriptional adaptation (Supplementary Fig. 9a and Supplementary Data 9). Epigenetic mechanisms such as DNA methylation are known to affect transcription levels of individual genes[17,18]. When we explored available DNA methylation data for a subset of samples from the TCGA dataset ($n = 9317$), we observed that for a subset of genes their TACNA expression levels correlated with their mean methylation levels ($r$ range = −0.65–0.45, Supplementary Fig. 9b). These results indicate that DNA methylation could be one of the underlying mechanisms driving the degree of transcriptional adaptation to CNAs.

The function of protein complexes might depend on the correct proportional levels of protein subunits[19]. When CNAs occur, cancer cells might transcriptionally adapt the mRNA expression of genes that encode protein subunits to maintain their ratios at the protein level. However, in the TCGA dataset we observed very little enrichment of genes having a high degree of transcriptional adaptation in 304 protein complexes with 5 or more members as defined by the CORUM protein complex definitions (see Supplementary Note 1 and Supplementary Data 10). We did observe for 18 out of 304 protein complexes (permutation $P$ value < 0.05) that patient-derived cancer cells might coordinately adjust the mRNA expression levels of some

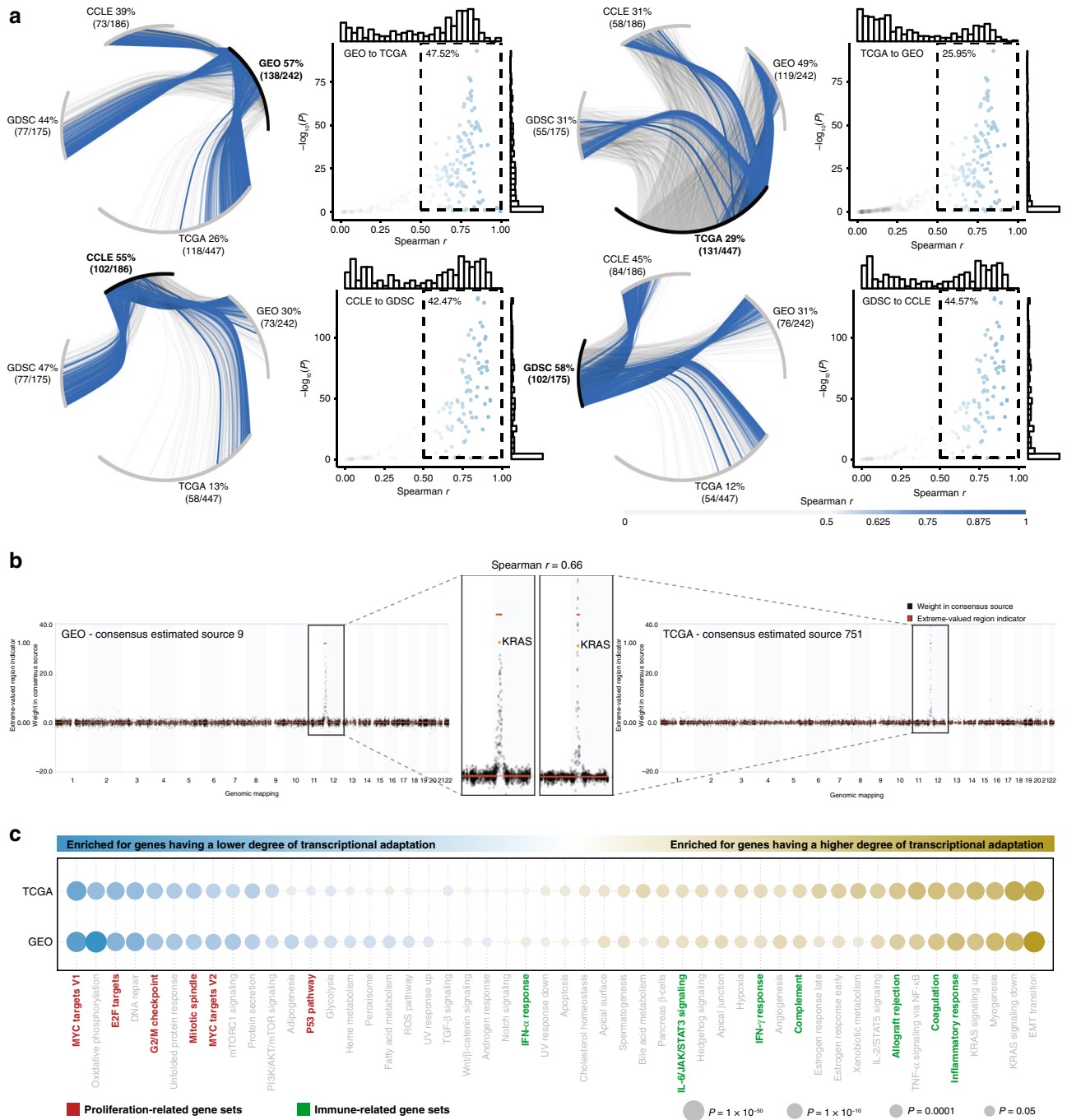

**Fig. 3 Degree of transcriptional adaptation to CNAs. a** Citrus plots showing the Spearman correlations between CNA-CESs in the reference dataset, depicted in bold, with CNA-CESs in the other three datasets. Blue lines indicate $r > 0.5$. Correlations are calculated based on the weights of genes in marked genomic regions of the CNA-CESs under investigation. Each scatterplot with marginal histograms shows the correlations versus their $-\log_{10}$ transformed $P$ values. The inset shows correlations $> 0.5$ having a $P$ value $< 0.05$. **b** Example of a CNA-CESs in the GEO dataset that is highly correlated with a CNA-CES in the TCGA dataset with *KRAS* having a low degree of transcriptional adaptation to CNAs. Spearman correlation coefficient was derived using the genes mapping to the extreme-valued region from either of the CNA-CESs ($n = 38$). **c** Enrichment results using two-sided Welch's $t$-test for the MSigDB Hallmark collection. A yellow bubble indicates enrichment for genes with a high degree of transcriptional adaptation to CNAs, and a blue bubble indicates a low degree. The size of the bubble corresponds to the significance level. Only CNA-CESs having at least 50 genes in their marked region were included. EMT: epithelial-mesenchymal transition; ROS: reactive oxygen species.

protein complex subunits in response to the occurrence of CNAs (Supplementary Fig. 10 and Supplementary Data 11).

**Most genes show moderate to high adaptation to CNAs.** In both the GEO and TCGA dataset, most genes mapped to a

marked genomic region in only one CNA-CES (Supplementary Data 12). The degree of transcriptional adaptation of these genes ($n = 7641$) highly correlated between the GEO and TCGA dataset (Fig. 4a, $r = 0.84$). Of these 7641 genes, 786 (10.2%) had low adaptation (metric $< 0.25$), 3898 (51.0%) moderate adaptation (metric 0.25–0.75), and 2957 genes (38.7%) high adaptation

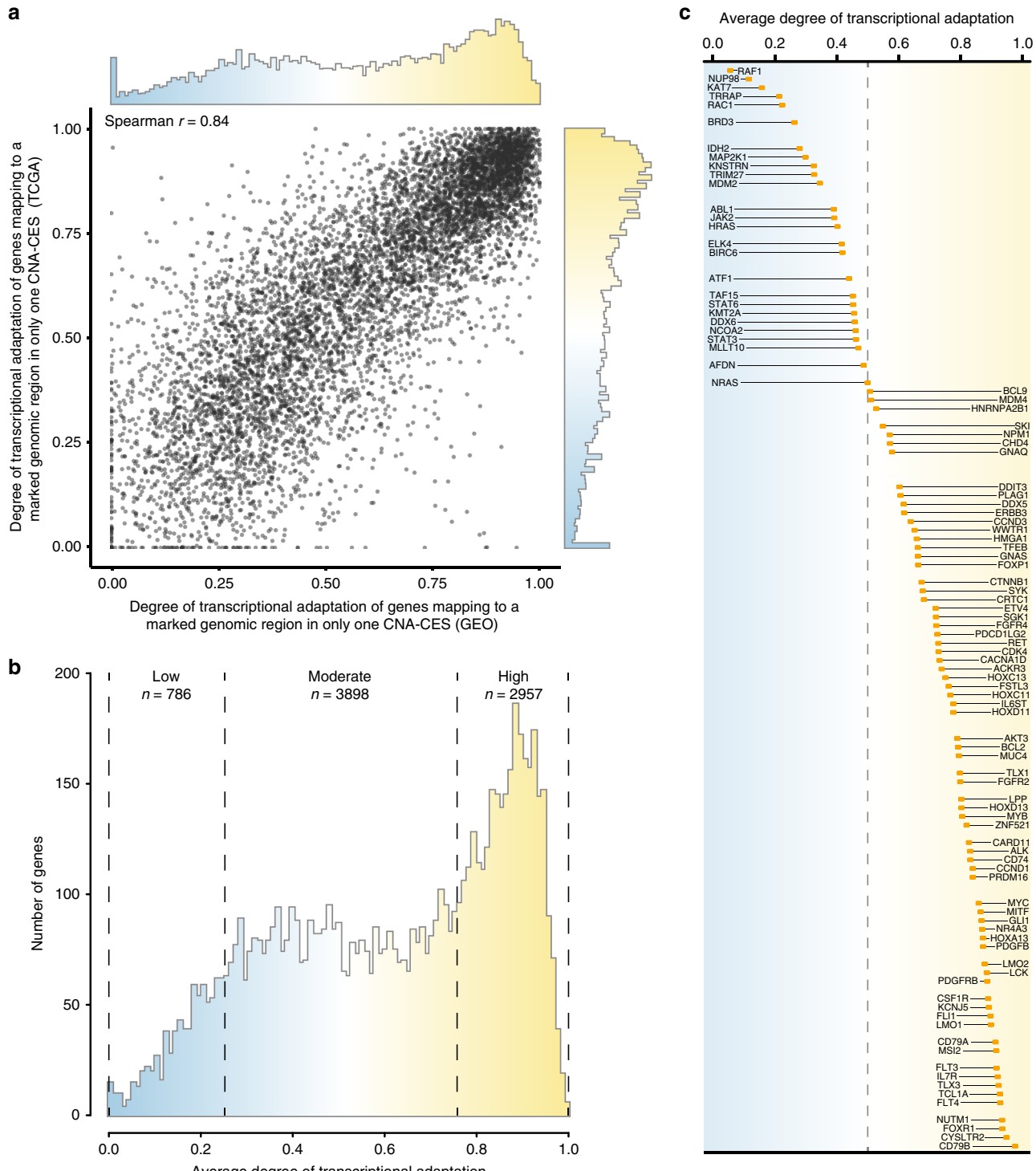

**Fig. 4 Degree of transcriptional adaptation to CNAs per individual gene. a** Degree of transcriptional adaptation to CNAs for individual genes mapping to a marked genomic region in a single CNA-CES in both the GEO and TCGA dataset ($n = 7641$). **b** Distribution of the average degree of transcriptional adaptation to CNAs for genes mapping to a marked genomic region in a single CNA-CES in both the GEO and TCGA dataset. **c** Average degree of transcriptional adaptation to CNA for a set of oncogenes obtained from the Catalogue of Somatic Mutations in Cancer Gene Census.

(metric > 0.75, Fig. 4b). These results suggest that for most genes the transcriptional effect of an underlying CNA at their genomic position is largely buffered by non-genetic adaptive mechanisms.

Amplification of oncogenes and the resulting enhanced gene expression levels have been implicated in the progression of many cancers[20]. We observed a large variability in the degree of

transcriptional adaptation in a set of oncogenes obtained from the Catalogue of Somatic Mutations in Cancer Gene Census (Fig. 4c and Supplementary Data 13)[21]. Whereas some oncogenes in this set had a low degree of transcriptional adaptation (e.g., *RAF1*, metric = 0.06), most showed a relatively high degree of transcriptional adaptation to CNAs (e.g., *MYC*, metric = 0.86).

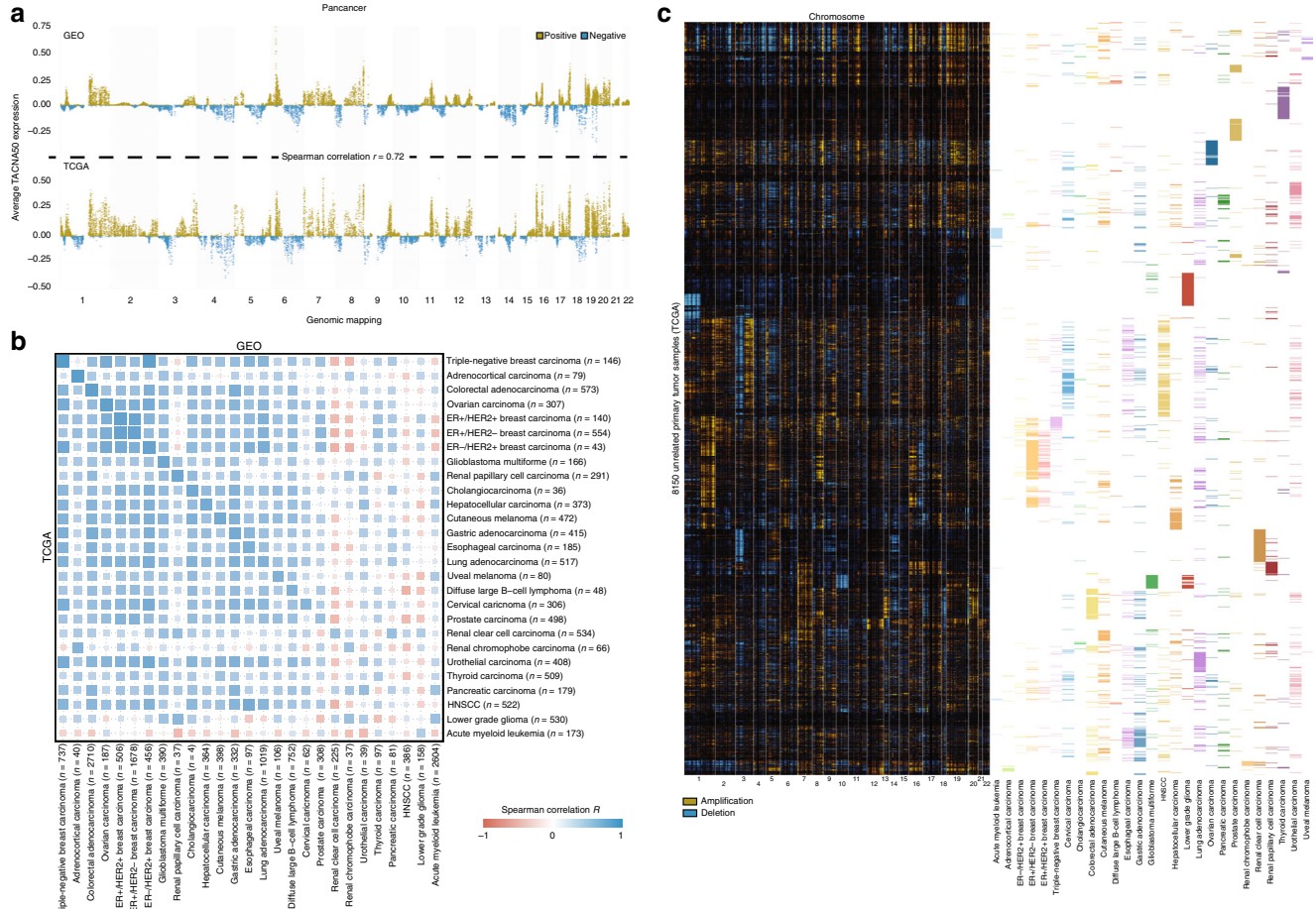

**Fig. 5 The landscape of transcriptional effects of CNAs in a large set of cancer samples. a** Average pan-cancer TACNA profiles in GEO and TCGA dataset. Spearman correlation coefficient was derived using the 15,389 genes which were present in both datasets. **b** Distance matrix of Spearman correlations between average TACNA profiles in the GEO dataset and TCGA dataset for overlapping tumor types. The size and transparency of a square corresponds to the absolute correlation coefficient. HNSCC: head and neck squamous cell carcinoma. **c** Hierarchical clustering of the landscape of transcriptional effects of CNAs in the TCGA dataset for 8150 cancer samples of tumor types also present in the GEO dataset.

This suggests that also transcription of many oncogenes is under the strong control of non-genetic regulatory factors that buffer the effects of CNAs.

**Transcriptional effects of CNAs in >21,000 cancer samples.** TACNA profiling was applied to samples from overlapping tumor types in the GEO dataset ($n = 13,180$) and TCGA dataset ($n = 8150$) to define the landscape of transcriptional effects of CNAs. This showed a high correlation between the average pan-cancer TACNA profiles in the GEO and TCGA dataset ($r = 0.73$, Fig. 5a). These average pan-cancer patterns of transcriptional effects of CNAs closely resemble average pan-cancer copy-number patterns reported in previous studies[22,23]. As expected, high correlations between the average GEO and TCGA TACNA profiles of copy-number-driven tumors such as triple-negative breast carcinoma ($r = 0.80$), colorectal adenocarcinoma ($r = 0.74$), and ovarian carcinoma ($r = 0.73$) were observed (Fig. 5b and Supplementary Data 14).

While exploring the landscape of transcriptional effects of CNAs, we observed common patterns consistent with well-known genomic alterations. For instance, many renal clear cell carcinomas in the TCGA dataset contained a pattern consistent with a deletion in chromosome 3p and an irregular amplification in chromosome 5q (Supplementary Fig. 11a)[24]. Likewise, we observed patterns consistent with a chromosome 10 deletion in

glioblastoma multiforme, and a 1p/19q deletion in low-grade gliomas (Supplementary Figs. 11b and c)[25,26]. The landscape of transcriptional effects of CNAs can be explored at www.genomicinstability.org (Supplementary Fig. 11d).

Hierarchical clustering of the landscape of transcriptional effects of CNAs in the GEO and TCGA dataset showed clear differences and commonalities between tumor types (Fig. 5c and Supplementary Figs. 12–21), indicating that tumor types share strong transcriptional effects of CNAs driving biological pathways relevant for their tumor behavior. The landscape of transcriptional effects of CNAs can enable transcriptomic-wide association studies (TWASs) to generate strong hypotheses on genes and biological pathways affected by CNAs that might be causally linked to tumor phenotypes.

**TACNA profiles are associated with immune composition.** Recently, a higher CNA burden was found to be negatively associated with an expression-based metric describing CD8+ T cell activity in the tumor microenvironment[23,27]. The ability to elicit an anti-cancer immune response with immune checkpoint inhibitors depends on the composition and activity of immune cells present in the tumor microenvironment, especially on the presence of CD8+ T cells[28]. However, it is still unclear how CNAs affect the activity and composition of immune cells present in the tumor microenvironment.

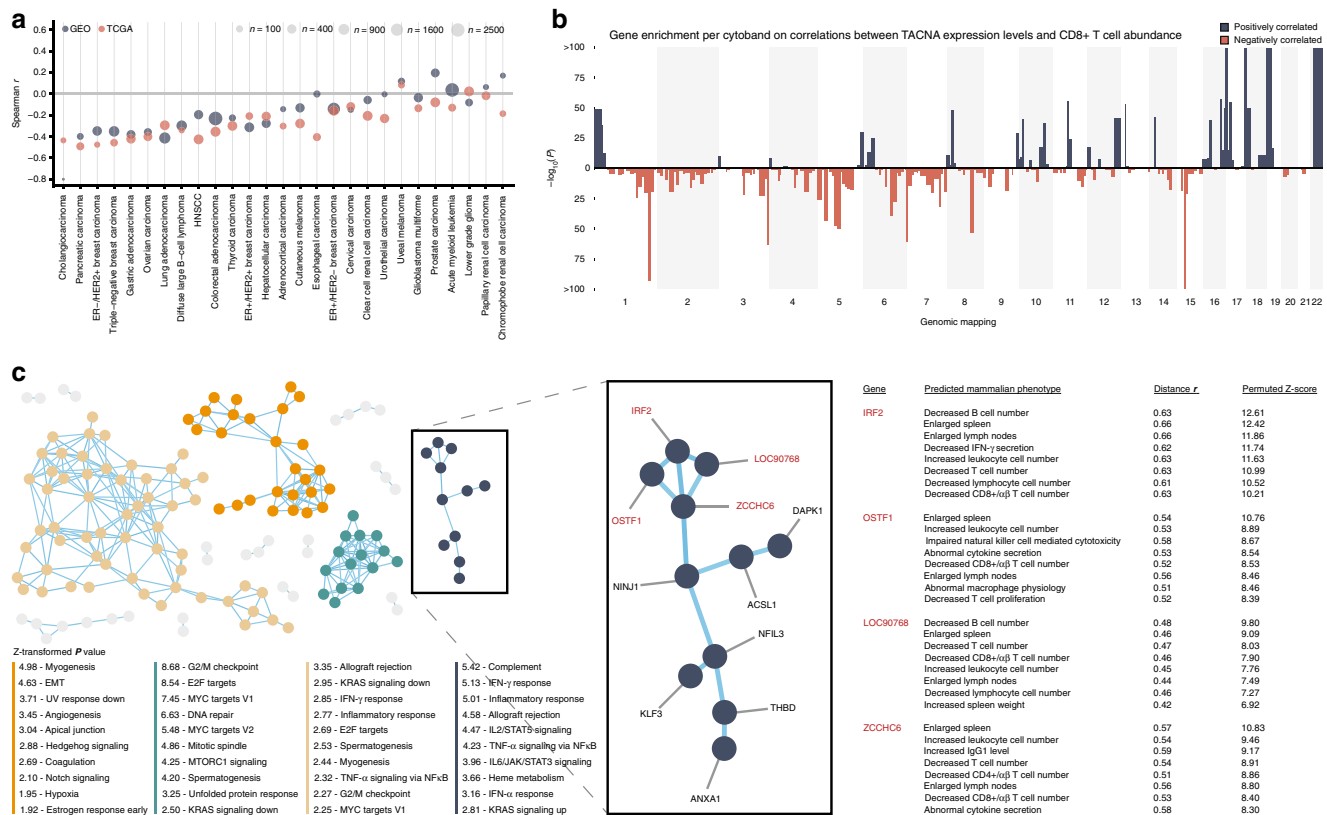

**Fig. 6 Transcriptional effects of CNAs in relation to inferred immunological phenotypes. a** Per tumor type, Spearman correlations between inferred CNA burden and an expression-based metric describing CD8+ T cell activity for each sample. **b** Manhattan plot showing, the enrichment (−log$_{10}$ P value on the y-axis) for genes with a strong correlation between TACNA expression level and inferred CD8+ T cell abundance per cytogenetic band defined according to the MSigDB Positional gene sets collection. **c** Left panel: constructed co-functionality network for the top 350 genes with the highest inverse correlation between their TACNA expression levels and CD8+ T cell abundance. Middle panel: cluster in which genes share strong predicted co-functionality (r > 0.8) within the co-functionality network that was enriched with genes predicted to be involved in immunological processes (e.g., complement, inflammatory response, and IFN-γ response). Right panel: a GBA strategy with >106,000 expression profiles in combination with the gene set collection obtained from the Mammalian Phenotype (MP) Ontology predicted for the genes IRF2, OSTF1, LOC90768, and ZCCHC6 that altered transcription results in a decreased CD8+/αβ T cell number.

We therefore developed a univariate measurement describing the CNA burden for each sample (see Methods), which confirmed the negative association between CNA burden and the expression-based metric describing CD8+ T cell activity in both the GEO and TCGA datasets (Fig. 6a, Supplementary Fig. 22 and Supplementary Data 15). We observed strong negative correlations especially in the prototypical copy-number-driven tumors. To determine how CNAs affect the composition of immune cells present in the tumor microenvironment, we inferred the abundance of 22 immune cell types with CIBERSORT for the 21,592 cancer samples in the GEO dataset and correlated these to TACNA expression levels (Supplementary Data 16). Genes were ranked based on this correlation and GSEA was performed with the MSigDB Positional gene sets collection. We identified multiple genomic regions that were enriched for genes having a highly significant correlation with CD8+ T cell abundance, indicating that CNAs occurring at these genomic regions are driving the associations (Fig. 6b and Supplementary Data 17).

To identify the genes in these enriched genomic regions that may be linked to CD8+ T cell abundance, we constructed a co-functionality network for the top 350 genes with highest inverse correlation between their TACNA expression levels and CD8+ T cell abundance. This co-functionality network was generated with an integrative tool that predicts gene functions based on a

guilt-by-association (GBA) strategy utilizing >106,000 expression profiles (see Methods). We identified four big clusters in which genes have strong predicted co-functionality (r > 0.8, Fig. 6c). One cluster was enriched with genes predicted to be involved in biological processes related to tumor progression (e.g., E2F targets, G2/M checkpoint, and MYC targets), whereas two clusters were enriched for immunological processes (e.g., allograft rejection, complement, and interferon-γ response). This suggests that CNAs can transcriptionally activate multiple processes that benefit tumor growth, in this case, simultaneously lowering CD8+ T cell abundance (i.e., avoiding immune destruction) and promoting tumor cell proliferation.

We then predicted the phenotypic effects of altered expression levels of the genes in the cluster with the strongest enrichment for immunological processes using the GBA strategy in combination with the gene set collection obtained from the Mammalian Phenotype Ontology. We predicted that altered expression levels of four genes (IRF2, OSTF1, LOC90768, and ZCCHC6) would indeed result in decreased CD8+/αβ T cell numbers (Fig. 6c).

The above data strongly suggest that the identified genes which are transcriptionally affected by CNAs underlie the composition and activity of immune cells present in the tumor microenvironment. These genes could potentially serve as novel targets to induce or enhance anti-cancer immune responses.

## Discussion

In the present study we revealed the degree of transcriptional adaptation to CNAs in a genome-wide fashion using 34,494 gene expression profiles. Using TACNA profiling, we defined the landscape of transcriptional effects of CNAs and determined how these effects might influence the composition of immune cells in the tumor microenvironment.

We observed that most oncogenes in our set had a high degree of transcriptional adaptation to CNAs. This suggests that elevation of expression levels of many oncogenes is only beneficial for tumor progression up to a certain level; excessive expression of several oncogenes was previously linked to cell senescence and apoptosis[29]. Perturbations of genes whose expression levels are tightly controlled by non-genetic factors (i.e., have a high degree of transcriptional adaptation to CNAs) could thus result in more extreme tumor phenotypes. These genes could therefore be potential therapeutic targets. In addition, our results facilitate future research into the underlying mechanisms of transcriptional adaptation of genes to CNAs, which are currently poorly understood[7,30].

TACNA profiling was applied to gene expression profiles in two patient-derived datasets and two cell line compendia. These expression profiles were generated with three platforms using either RNA-seq or microarray technology. TACNA profiling proved robust, as shown by the highly reproducible average pan-cancer and tumor-type-specific TACNA profiles in the GEO and TCGA datasets. This enables reanalysis of the tens of thousands publicly available gene expression profiles generated from various platforms that are lacking a paired CNA profile. This approach provides a novel tool to determine how CNAs drive the progressive acquisition of tumor phenotypes, such as the hallmarks of cancer[29].

After generating the landscape of transcriptional effects of CNAs, we identified four genes which are associated with reduced CD8+ T cell abundance in the tumor microenvironment. This is especially of interest as a high number of CD8+ T cells is associated with higher response rates to immune checkpoint inhibitors[31–33]. Targeting these potentially causal genes might increase CD8+ T cell infiltration. In combination with immune checkpoint inhibitors, this might induce or enhance an anti-cancer immune response, and thereby improve disease outcome.

In conclusion, our study provides a new tool to explore the transcriptional effects of CNAs which can lead to novel biological insights into how CNAs drive tumor behavior, and guide the development of new therapeutic strategies.

**URLs**. The degree of transcriptional adaptation, source code of TACNA profiling, landscapes of transcriptional effects of CNAs per dataset, and additional datasets and results are available at http://www.genomicinstability.org/.

## Methods

**Data acquisition**. Microarray expression data was collected from three public data repositories: GEO platform GPL570 (generated with Affymetrix HG-U133 Plus 2.0)[34], CCLE (generated with Affymetrix HG-U133 Plus 2.0)[35], and GDSC (generated with Affymetrix HG-U219)[36]. Preprocessing and aggregation of raw expression data was performed using the robust multi-array average algorithm with RMAExpress (version 1.1.0)[37]. Quality control of the processed expression data is described in the Supplementary Note 1. Pre-processed and normalized RNA-seq data was collected from TCGA using the Broad GDAC Firehose portal. In each dataset, expression levels for every gene were standardized to a mean of zero and variance of one. In addition, CNA profiles generated with Affymetrix Genome-Wide Human SNP Array 6.0 were collected for a subset of samples in the TCGA dataset, CCLE dataset and GDSC dataset and processed as described in Supplementary Note 1.

**Identification of CNA-CESs**. ICA was used to identify a regulatory model for the mRNA transcriptome[12]. For each dataset containing mRNA expression profiles of

$p$ genes from $n$ samples, ICA resulted in (i) the extraction of $i$ independent components (hereafter called estimated sources, ESs), (ii) an ES matrix of dimension $i \times p$, where the weight of each ES represents the direction and magnitude of its effect on the expression level of each gene, and (iii) a mixing matrix (MM) of dimension $i \times n$ which contains the coefficients (i.e., indirect activity measurements) of ESs in each sample. Within each dataset, principal component analysis was performed on the covariance matrix between samples, after which $i$ was chosen as the minimum number of top principal components which captured at least 85% of the total variance in the dataset. The inner product between the vector of coefficients in the MM and the vector of ES weights per individual gene results in the original mRNA expression level.

In ICA, an initial random weight vector with a variance of 1 has to be chosen in order to obtain statistically independent ESs. Hence, varying initial random weight vectors could result in different sets of ESs. To retrieve a set of consensus ESs (or CESs), we performed 25 ICA runs, each with a different random initialization weight vector. After all ICA runs were completed, ESs with an absolute Pearson $r > 0.9$ were clustered, where the number of ESs in each cluster could be at most the total number of ICA runs. CESs were computed by obtaining the average of ESs inside each cluster. Next, we calculated a credibility index for each CES by determining the ratio between the number of ESs in its cluster and the total number of ICA runs (i.e., 25). CESs with a credibility index of ≥50% were used for obtaining the CES matrix and the consensus MM. We hypothesized that each CES describes the effect of a latent transcriptional regulatory factor on gene expression levels.

A subset of CESs harbored a pattern in which genes having high absolute weights mapped to a specific contiguous genomic region (i.e., CNA-CESs). To identify these CNA-CESs, we developed and used a detection algorithm, which is described in detail in the Supplementary Note 1.

**Transcriptional adaptation to CNA profiles**. For each dataset, weights of genes mapping to a contiguous genomic region in CNA-CESs marked by the detection algorithm were retained in the CES matrix. Weights of genes that mapped outside marked genomic regions were instead set to zero. Next, TACNA profiles were calculated as the product between this transformed CES matrix and the consensus MM. TACNA profiles in the GEO dataset and TCGA dataset were adjusted by subtracting the Hodges Lehmann estimate (i.e., robust mean) of the TACNA expression of genes observed in normal tissue samples in each dataset. This ensured that a TACNA expression value of zero corresponded to a situation where a gene has exactly two copies on the genome.

**Human fragile sites**. Previously described genomic locations of aphidicolin-induced fragile sites identified through cytogenic analyses were used to assess the colocalization of borders of marked genomic regions in CNA-CESs with common fragile sites[15]. Genomic coordinates were converted to human genome reference GRCh38 using the Batch Coordinate Conversion tool from USCS Genome Browser[38]. Borders of marked genomic regions in CNA-CESs were assumed to colocalize with common fragile sites when one or both borders were located inside a common fragile site.

**Gene set enrichment analyses**. Weights of genes in a marked genomic region of a CNA-CES were transformed to metrics ranging from zero to one, where zero would correspond to the highest absolute weight (i.e., low degree of transcriptional adaptation to CNAs) and one would correspond to the lowest absolute weight (i.e., high degree of transcriptional adaptation to CNAs). When genes occurred in marked regions of multiple CNA-CESs, the lowest metric was used. Enrichment of gene sets was tested according to the two-sample Welch's $t$-test for unequal variance. The proportion of false discoveries was controlled using a multivariate permutation testing framework with 100 permutations, a 1% false discovery rate and confidence level of 80%.

**Inferred CNA burden**. For each sample in the GEO dataset and TCGA dataset, CNA load was estimated as the sum of the coefficients (i.e., indirect activity measurements) of all CNA-CESs in the MM with at least 50 genes in their marked genomic regions. CNA loads were normalized to a 0–1 range across samples of the same tumor type in each dataset separately.

**Expression-based immune metric**. A previously defined set of genes describing CD8+ T cell and natural killer cell activity (*CD2, CD3E, CD247, GZMK, NKG7,* and *PRF1*) was used to calculate immune metric scores[27]. Per sample, the rank position of the mRNA expression levels of each of these genes was calculated. Scores for each sample were determined by calculating the mean rank position of the seven genes. Immune metric scores were normalized to a 0–1 range across samples of the same tumor type in each dataset separately.

**Inference of immune cell type abundance**. Inference of the abundance of 22 immune cell types was performed using CIBERSORT[39] and confined to the GEO dataset. As CNAs generally do not occur in non-tumor tissue, we reconstructed gene expression profiles using all CESs except CNA-CESs with at least 50 genes in

their marked regions to more accurately capture the effects from the tumor microenvironment on gene expression levels. Next, the abundance of 22 immune cell types was inferred by applying the leukocyte gene signature matrix (LM22) on the reconstructed profiles.

**Prediction of gene functionalities**. We used a GBA approach to predict likely functions for genes based on gene co-regulation. For this, we conducted a consensus-ICA on an unprecedented scale (manuscript in preparation). In short, a covariance matrix was calculated between 19,635 genes using the expression patterns of 106,462 gene expression profiles generated with Affymetrix HG-U133 Plus 2.0 representing the many disease states, cellular states, and genetic and chemical perturbations that were obtained. Consensus-ICA was performed on the covariance matrix, which resulted in the identification of a large set of CESs and a MM reflecting the activity of each source in the expression pattern of each gene across the samples. Next, a GBA approach was used to predict the functionality of individual genes. First, we retrieved 16 public gene set collections describing a large range of biological processes and phenotypes. For each gene set, we calculated its bar code by averaging the MM weights of its member genes per CES. Next, for each gene in the MM, the distance correlation was determined between its MM weights and the gene set bar code. A high correlation between a gene's MM weight and a gene set bar code indicated that the gene under investigation shared a functionality with the genes of the specific gene set under investigation. Significance levels were obtained with permutated data (1000 permutations). This strategy was used on 23,372 well-described functional gene sets, which enabled us to create a comprehensive network of predicted functionalities of individual genes. This framework is available at http://www.genetica-network.com.

A more detailed description of the methods used in this study is provided in the Supplementary Note 1.

**Reporting summary**. Further information on research design is available in the Nature Research Reporting Summary linked to this article.

## Data availability

Microarray expression data was collected from three public data repositories: Gene Expression Omnibus with accession number GPL570 (generated with Affymetrix HG-U133 Plus 2.0), CCLE (generated with Affymetrix HG-U133 Plus 2.0, file CCLE_Expression.Arrays_2013-03-18.tar.gz) available at https://portals.broadinstitute.org/ccle/data and GDSC (generated with Affymetrix HG-U219) available at https://www.ebi.ac.uk/arrayexpress/experiments/E-MTAB-3610/. Pre-processed and normalized RNA-seq data was collected from TCGA using the Broad GDAC Firehose portal (https://gdac.broadinstitute.org/). Individual identifiers or accession numbers for all samples used in this paper can be found in the Download support data section of http://www.genomicinstability.org/. The datasets generated during and/or analyzed during the current study are available in the website http://www.genomicinstability.org/. TACNA gene distribution and degree of TACNA can be explored at the gene level in top four panels of the above website. In addition, the tab Download support data in the website links to zip files that contain TACNA profiles and degree of TACNA estimates for all four datasets.

## Code availability

The website http://www.genomicinstability.org/ contains a link to the R code that contains the pipeline used to calculate the TACNA profiles.

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

## Acknowledgements

This research was supported by the Netherlands Organization for Scientific Research (NWO-VENI grant 916-16025 to R.S.N.F.), the Dutch Cancer Society (RUG 2013-5960 to R.S.N.F. and RUG 2016-10034 to E.G.E.d.V.), the European Research Council (ERC Consolidator grant 682421 to M.A.T.M.v.V.), and the Graduate School of Medical Sciences of the University Medical Center Groningen (no grant number, to R.D.B.).

## Author contributions

R.S.N.F. conceived this study. A.B., R.D.B., and R.S.N.F. collected and assembled data. A.B., R.D.B., C.G.U., and R.S.N.F. performed data analyses. A.B., R.D.B., C.G.U., E.G.E.d.V., M.A.T.M.v.V., and R.S.N.F. contributed to the data interpretation, writing of the paper, and the final decision to submit the paper.

## Competing interests

E.G.E.d.V. reports institutional financial support for advisory board/consultancy from Sanofi, Daiichi, Sankyo, NSABP, Pfizer and Merck, and institutional financial support for clinical trials or contracted research from Amgen, Genentech, Roche, AstraZeneca, Synthon, Nordic Nanovector, G1 Therapeutics, Bayer, Chugai Pharma, CytomX Therapeutics and Radius Health, all unrelated to the submitted work. All other authors declare no competing interests.
