## [Peer Review File · Nature Communications]

Reviewers' comments:

Reviewer #1 (Remarks to the Author):

Structural chromosomal aberrations can result in DNA copy-number alterations (CNAs) that reduces the expression of genes within segments that are deleted, and conversely, increase the expression of genes when they are amplified. However, changes in gene expression do not always scale proportionally with changes in DNA content, suggesting that the abundance of specific genes may be controlled through mechanisms beyond simple gene dosage effects. Bhattacharya et al. developed a computational method called “transcriptional adaptation to CNA profiling” (TACNA profiling) to accurately infer CNAs from transcriptomic data derived from RNA sequencing and microarrays without the need for DNA content information. The authors analyzed a large number of samples (34,494) made available from public repositories and identified cohorts of genes whose expression patterns failed to correlate with DNA copy-number. The authors propose that these “transcriptionally adapted” genes are associated with specific biological processes: for example, low adaptation for genes involved in proliferation and high adaptation for immune-related genes.

The manuscript is well written, easy to follow, and the data are nicely presented. Overall, this is an impressive study that I recommend for publication in Nature Communications pending minor experimental revisions. I only have a few questions/suggestions for improvement.

1) The authors propose that TACNA profiling is platform-independent. The authors should analyze a cohort of samples that have been subjected to both RNA sequencing and microarray to determine whether there is a clear correlation between the two platforms. Along those lines, a validation cohort in which both genomic and transcriptomic information are available should be used to confirm that CNAs identified by DNA sequencing or SNP analyses are highly correlative with the proposed CNAs identified by TACNA profiling.

2) The authors should determine whether tissue/cells/DNA/RNA are available from some samples to confirm their results experimentally, for example, by analyzing samples with predicted CNA/gene expression changes by DNA FISH and quantitative PCR, respectively. Additionally, several genes exhibiting low and high adaptation should be tested using a similar method. If these reagents are not available, the authors could subject RNA sequencing data from human cell lines with known karyotypes to TACNA profiling.

3) Experimental systems of aneuploidy show that while the expression of most genes follows a general pattern across the extra trisomic chromosome, some genes clearly do not follow this trend

(PMID: 15466185 and 22968442 to name a few). The authors should investigate whether these genes fall within their subgroups for transcriptional adaptation. Along those lines, do genes that exhibit high adaptation encode components of protein complexes whose function, activity, or stability could be dysregulated by the stoichiometric imbalances of individual subunits?

5) Is the degree of adaptation related to the position or locus of the gene? For example, genes harbored within euchromatin vs. heterochromatin? How do adaptation profiles change with tissue/cancer type?

6) Line 126: The authors propose that a proportion of CNAs colocalized within common fragile sites, yet the location of most common fragile sites have only been mapped using low resolution methods. This should be further addressed or discussed.

Reviewer #2 (Remarks to the Author):

Overview

Tumor pathogenesis and continued oncogenesis is thought to be largely driven by perturbed regulatory pathways at key genes.

CNAs are known to dramatically alter the abundance of nearby genes, and thus make prime candidates for studying the effects on

tumor development. Characterizing the effect of CNAs on tumor expression requires genetic + expression assays in large numbers

of individuals in order to have enough statistical power to overcome experimental design artifacts (tissue heterogeneity) and

technical artifacts (sequencing/library bias). Bhattacharya and Bense et al describe TACNAS, an approach to profile the effects

of CNA on gene expression using a single gene expression profile.

Main Comments

1. The ICA and TACNAS approaches are described in detail in the supplementary note with an accompanying figure. However it would be very

helpful to give an exposition of the approach and a high-level overview at the start of the main text/results. Considering this

represents the major advancement, along with its findings, it would set the tone for the rest of the article as well as place

things in a more interpretable context.

2. The authors demonstrate that the identified CNAs predicted from the TACNAS approach are highly correlated on average with actual CNA

measured within the same dataset. Could the authors also show that this finding is robust to a cross-validation approach? I suggest this

be done in two ways.

The first would be to perform k-fold cross-validation within a dataset. That is, hold out a fold, fit the ICA model to compute weights and

predict on the held out set. Repeat for the remaining folds and then compute correlation with actual CNAs. It should be noted that the

matrix whitening here should not include the held-out set.

Similarly, while the authors showed that -weights- are correlated for some genes across studies, it would be very convincing if the authors

could also perform a cross-study cross-validation. For example, take the weights inferred from TCGA and apply to CCLE mixing matrix and then

compute the correlation with actual CNAs. Repeating this for all possible pairs would help get a strong sense of the generalizability of

this approach.

3. The authors take care to clean and prepare the data as much as possible, but it is not clear how various relevant covariates that should

drive large heterogeneity into account. For example, biological sex, individual's age, tumor tissue type, etc.

4. For genes identified using TACNAS where correspond data have actual CNA or imputed CNA from genotype, can the authors perform a QTL analysis

while controlling for relevant variables? There was some work showing that CNAs exist in more extreme regions, but it would be informative

to see that actual CNA variation is statistically linked with expression values at these regions.

Referee #1

We are delighted that the referee finds our manuscript is well-written, easy to follow and that our data are nicely presented. The following suggestions for improvement were made:

Remark 1: "The authors should analyze a cohort of samples that have been subjected to both RNA sequencing and microarray to determine whether there is a clear correlation between the two platforms."

To address this remark, we collected a set of samples ($n = 570$) from The Cancer Genome Atlas (TCGA) for which an expression profile was generated with both RNA sequencing and microarray profiling (Affymetrix HG-U133A). These samples represent 295 serous ovarian cancers, 130 lung squamous cell cancers and 145 glioblastoma multiforme tumors. Next, we independently applied TACNA profiling on both the RNA sequencing and microarray profiles. Although the two techniques used to generate these profiles differ substantially, we observed a high correlation (distribution mode $r = 0.66$) between the paired TACNA profiles (see panel a of figure below). The correlations between these RNA sequencing and microarray derived TACNA

profiles and their paired CNA profiles (derived from SNP arrays) had a distribution mode with $r = 0.62$ and $r = 0.54$, respectively (see panels b and c of figure below). The lower correlations between CNA profiles and TACNA profiles observed in this set of 570 samples could be explained by the fact that 2 out of the 3 tumor types (lung squamous cell cancer and glioblastoma multiforme) belong to the more 'point mutation-driven' tumor types which contain a limited number of CNAs (see panel d of figure below). Moreover, the genes used for calculating the correlations in this analysis were limited to those shared by the RNA sequencing and microarray platforms. In addition to the results already provided in the previous version of the manuscript, this analysis further supports our claim that TACNA profiling can be performed on both platforms and is able to extract similar patterns from their gene expression profiles. To highlight these results, we added the following graphs as a new supplementary figure 5.

Supplementary Fig. 5. **a** Pearson correlations between paired TACNA profiles derived from RNA sequencing and microarray profiles ($n = 570$). **b** Distribution of Pearson correlations between TACNA profiles derived from RNA sequencing data and paired CNA profiles. **c** Distribution of Pearson correlations between TACNA profiles derived from microarray data and paired CNA profiles. **d** Variance observed in CNA profiles versus Pearson correlations between TACNA profiles derived from RNA sequencing data and microarray data.

The following text was added to the results section of our manuscript (page 9, lines 174-176):

“Moreover, we performed TACNA profiling using 570 samples from the TCGA that were subjected to both RNA sequencing and microarray profiling (Affymetrix HG-U133A). Paired TACNA profiles generated with RNA-seq and microarray data were highly correlated (Supplementary Fig. 5, mode of distribution $r = 0.66$).”

Remark 2: “A validation cohort in which both genomic and transcriptomic information are available should be used to confirm that CNAs identified by DNA sequencing or SNP analyses are highly correlative with the proposed CNAs identified by TACNA profiling.”

We absolutely agree with the referee that CNAs identified in genomic profiles are expected to be highly correlated with patterns observed in paired TACNA profiles. In line with this notion, we demonstrated that TACNA profiles strongly correlate with their paired, independently generated genomic profiles. These findings are described in the results paragraph “TACNA profiles highly correlate with SNP-derived profiles”. Specifically, in Fig. 2a we show per dataset examples of how patterns in TACNA profiles match those in paired CNA profiles. In Fig. 2b, we show the distribution of correlations between TACNA profiles and their paired genomic profiles for samples from the TCGA ($n = 10,620$), CCLE ($n = 1,011$), and GDSC ($n = 962$) datasets. We believe these results are already fully in line with the suggestion made by the referee.

Remark 3: “The authors should determine whether tissue/cells/DNA/RNA are available from some samples to confirm their results experimentally, for example, by analyzing samples with predicted CNA/gene expression changes by DNA FISH and quantitative PCR, respectively. Additionally, several genes exhibiting low and high adaptation should be tested using a similar method. If these reagents are not available, the authors could subject RNA sequencing data from human cell lines with known karyotypes to TACNA profiling.”

As described in the answer to remark 2, our previous manuscript already contained an analysis confirming high correlations between TACNA profiles and paired CNA profiles (independently generated with a SNP array) for a total of 12,593 samples, some of which originate from cell lines. However, as this analysis was only performed using samples from the TCGA, CCLE and GDSC datasets, we therefore explored patient-derived acute myeloid leukemia samples in the GEO dataset for which patient karyotype information was provided online by the original authors ($n = 189$). As observed with the other datasets, we also observed in these GEO samples clearly matching patterns between TACNA profiles and the expected karyotype.

In addition, we performed expression quantitative trait loci (eQTL) analyses (see response to remark 13) using the mRNA expression profiles, TACNA profiles and CNA profiles (derived from SNP arrays) from the TCGA dataset ($n = 10,620$). From this analysis we observed a clear association between the degree of TACNA and the correlation between CNAs and regular mRNA levels (Spearman $r = -0.83$). For genes with a high degree of transcriptional adaptation we observed low correlation. In contrast, for genes with a low degree of transcriptional adaptation we observed higher correlations between CNAs and regular mRNA levels.

We added the following supplementary figure to our manuscript to highlight several examples of the TACNA profiles of these acute myeloid leukemia samples:

Supplementary Fig. 2 Examples of matching patterns between TACNA profiles and the expected karyotype in acute myeloid leukemia samples in the GEO dataset.

The following text was added to the manuscript (page 7, lines 129-132):

“In a subset of patient-derived acute myeloid leukemia samples in the GEO dataset for which karyotype information was provided online ($n = 189$) we also observed clearly matching

patterns between TACNA profiles and the expected karyotype in these samples (Supplementary Fig. 2).”

Remark 4: “Experimental systems of aneuploidy show that while the expression of most genes follows a general pattern across the extra trisomic chromosome, some genes clearly do not follow this trend (PMID: 15466185 and 22968442 to name a few). The authors should investigate whether these genes fall within their subgroups for transcriptional adaptation.”

To address this remark, we downloaded expression profiles of an experimental HCT116 cell line model of aneuploidy as reported by Stingele et al.¹ Differences between the mRNA levels of parental HCT116 cell lines (replicates = 3) and an isogenic model with chromosome 5 tetrasomy (replicates = 3) were calculated (see panel a of figure below). The same differences were calculated using TACNA profiles generated for these 6 samples with CNA-CESs identified within the TCGA data (see panel b of figure below and Cross-study cross-validation section in Supplementary Note for methods). For genes located on chromosome 5, we observed a larger signal-to-noise ratio in differences of TACNA expression levels compared to regular mRNA expression levels (see panel c of figure below). We would like to point the referee to our response to remark 13 for which we performed this type of analysis on a larger scale by eQTL analyses using the mRNA expression profiles, TACNA profiles and CNA profiles (derived from SNP arrays) from the TCGA dataset ($n = 10,620$). Both analyses indicate that TACNA profiling increases the statistical power to detect eQTLs, i.e. associations between CNAs and mRNA expression levels.

We did not observe a significant correlation between the differences in TACNA expression levels and differences in regular mRNA expression levels of individual genes on chromosome 5. We previously have shown that the effects of CNAs on gene expression levels are often overshadowed by experimental and other non-genetic factors.² Therefore, the effect of the induced chromosome 5 tetrasomy on mRNA expression levels may be small and overshadowed by other (non-genetic) effects that are not associated with specific single CNAs. For example, Stingele et al. showed that aneuploidy invokes a general cellular mRNA response which is not the result of any specific CNAs but only the presence of aneuploidy. This response is not captured as a CNA-CES because it is not the effect of any specific CNA and therefore is not present in the TACNA profile. Another possibility is that the small number of samples used in the experimental model of aneuploidy results in large confidence intervals of the observed \log_2 fold change in mRNA expression. Finally, we obtained the degree of transcriptional adaptation of genes to CNAs from patient-derived tumor gene expression profiles. *In vivo*, tumors cells develop transcriptional adaptive mechanisms under selective pressure to survive, whereas this selective pressure might be absent in *in vitro* models as used by Stingele et al.

Given these results, we added the following supplementary figure to our manuscript:

Supplementary Fig. 3 **a** Differences in expression levels between average TACNA profiles of three aneuploid samples and average TACNA profiles of three diploid samples. **b** Differences in expression levels between average standardized mRNA expression profiles of three aneuploid samples and average standardized mRNA expression profiles of three diploid samples. **c** Left: distribution of the differences in expression levels between average standardized mRNA expression of aneuploid and diploid samples on gene-level separately for chromosome 5 and other chromosomes. Middle: distribution of the differences in expression levels between average TACNA expression of aneuploid and diploid samples on gene-level separately for chromosome 5 and other chromosomes. Right: comparison between average differences in TACNA expression levels and average differences in standardized mRNA expression levels for chromosome 5.

We added the following text to the manuscript (pages 7-8, lines 133-146):

“We obtained gene expression profiles of an experimental model of aneuploidy introduced in HCT116 cells¹⁴. Differences on the mRNA level and TACNA level were calculated between the parental HCT116 cells ($n = 3$) and an isogenic model with chromosome 5 tetrasomy ($n = 3$) (Supplementary Fig. 3a and 3b). For genes located on chromosome 5, we observed larger signal to noise ratio in differences of TACNA expression levels compared to regular mRNA expression levels (Supplementary Fig. 3c). We did not observe a correlation between the \log_2 fold change in mRNA expression of individual genes and changes in TACNA expression levels ($r = 0.04$). We have previously shown that the effects of CNAs on gene expression levels are often overshadowed by other (non-genetic) effects that are not associated with specific single CNAs⁹. For example, Stingele et al. showed that aneuploidy invokes a general cellular mRNA response which is not the result of any specific CNAs but only the presence of aneuploidy¹⁴. Additionally, tumor cells *in vivo* may develop transcriptional adaptive mechanisms to survive under selective pressure, which may be different from the adaptive mechanisms observed *in vitro*.”

Based on the above analysis, we added the following paragraph to the Supplementary Note (page 46):

“TACNA profile versus mRNA changes upon chromosome 5 transfer

Expression profiles were downloaded from the Gene Expression Omnibus for samples GSM978891-96 belonging to series GSE39768 performed on platform GPL4133. In addition, 2,949 additional samples were downloaded for other studies performed on the same platform which had also Cy3 single label data available. All genes in this joint GPL4133 dataset were then median centered and \log_2 transformed. TACNA profiles were generated for the GPL4133 dataset using the sources obtained for the TCGA dataset (see Cross-study cross-validation section of Supplementary Note). To calculate arithmetic differences between the two conditions an average was calculated for the control arm (GSM978891-93) and another for the

experimental arm (GSM978894-96) of the HCT116 model of chromosome 5 tetrasomy performed in GSE39768. Pearson correlations were calculated between the differences in mRNA expression and the differences of TACNA profiles using only genes mapping to chromosome 5.”

Remark 5: “Do genes that exhibit high adaptation encode components of protein complexes whose function, activity, or stability could be dysregulated by the stoichiometric imbalances of individual subunits?”

It is reported that the function of protein complexes is dependent on the correct proportional levels of protein subunits³. When CNAs occur, cancer cells might transcriptionally adapt the mRNA expression of genes that encode protein subunits to maintain their ratios at the protein level. To explore the question raised by the referee, we collected gene sets from the CORUM gene set collection describing human protein complexes with 5 or more members.⁴ It could be the case that genes belonging to such a protein complex show a higher degree of transcriptional adaptation on average. We performed gene set enrichment analysis with the CORUM gene sets on the gene list ranked on the degree of transcriptional adaptation to CNAs. We observed very little enrichment of genes having a high degree of transcriptional adaptation in these protein complexes (Supplementary Table 10). This limited enrichment is in line with previous reports that show that dose adjustment of genes encoding protein complexes occurs mostly at the post-translational and not at the transcriptional level.¹

It could be that the degree of transcriptional adaptation to CNAs is enough to keep the ratio between genes part of a protein complex balanced without restoring the mRNA expression to the level observed in the normal diploid state. To explore this hypothesis, we created distance matrices (Pearson correlation) between all members of a protein complex using the TACNA expression value observed for each gene over all the samples in the TCGA dataset. Not taking into account the matrix diagonal, from this matrix we calculated the absolute median pairwise correlation as a density metric of the magnitude of internal correlation among members of a specific protein complex. A density metric of 1 signifies the scenario where all member genes have an identical pattern of mRNA expression levels due to underlying CNAs. In this scenario the ratio between TACNA expression levels for all genes part of a protein complex are kept consistent. To calculate the significance of the density metrics we performed a permutation test with 1,000 permutations. Out of 304 evaluated protein complexes 18 had a significant density metric ($P < 0.05$, Supplementary Table 11). As an example, the spliceosome complex, which comprises 130 genes had two subgroups of subunits with coordinated TACNA expression values over many samples in the TCGA dataset (see figure below). These results suggest that patient-derived cancer cells might coordinately adjust the

mRNA expression levels of some protein complex subunits in response to the occurrence of CNAs.

We added the following supplementary figure to our manuscript:

Supplementary Fig. 10 a Hierarchical clustering of the correlation distance matrix of genes belonging to the Spliceosome complex (CORUM) in the TCGA-dataset. b Hierarchical clustering of the transcriptional effects of CNAs in the TCGA dataset for genes belonging to the Spliceosome complex (CORUM) for 10,817 samples.

Given these results, the following text was added to the manuscript (page 11, lines 211-220):

“The function of protein complexes might depend on the correct proportional levels of protein subunits¹⁹. When CNAs occur, cancer cells might transcriptionally adapt the mRNA expression of genes that encode protein subunits to maintain their ratios at the protein level. However, in the TCGA dataset we observed very little enrichment of genes having a high degree of transcriptional adaptation in 304 protein complexes with 5 or more members as defined by the CORUM protein complex definitions (see Supplementary Note and Supplementary Table 10). We did observe for 18 out of 304 protein complexes (permutation P value < 0.05) that patient-derived cancer cells might coordinately adjust the mRNA expression levels of some protein complex subunits in response to the occurrence of CNAs (Supplementary Fig. 10 and Supplementary Table 11).”

We added the following text to the Supplementary Note (page 45):

“Protein complexes analysis

CORUM complex gene sets were collected from CORUM website ‘<https://mips.helmholtz-muenchen.de/corum/#download>’ (Core complex set). Complexes not mapping to ‘Human’ organism were discarded. Genes that had a ‘None’ value in the ‘subunits(Entrez IDs)’ column were discarded, which correspond to genes that have a UNIPROTID but no Entrez ID. Finally, protein complex gene sets with 4 or less subunits were discarded. Pearson correlation matrices for each protein complex ($n = 304$) were calculated using TACNA50 expression levels from the TCGA dataset ($n = 10,817$). For each correlation matrix, a density metric was defined as the median value of the correlation matrix after discarding the diagonal. For each protein complex, 1,000 similarly sized random groups of genes were generated and their density metric was calculated. A P value was assigned to each complex by calculating the fraction of permutations with lower density metrics as the real protein complex.”

Remark 6: “Is the degree of adaptation related to the position or locus of the gene? For example, genes harbored within euchromatin vs. heterochromatin?”

To explore whether the degree of transcriptional adaptation of genes is related to their genomic position, we performed gene set enrichment analysis on the degree of transcriptional adaptation using the MSigDB Positional Gene Sets collection.⁵ We observed that multiple genomic regions were enriched for genes having a higher or lower degree of transcriptional adaptation (see panel a of figure below). Furthermore, epigenetic mechanisms, including DNA methylation, have been shown to be involved in heterochromatin formation and affect gene transcription.^{6,7} We analyzed preprocessed methylation data generated with the Illumina 450 K array, which is available for a large subset of samples in the TCGA dataset ($n = 9,317$). For each sample, we obtained the β -values of all individual genes. These β -values for individual genes were calculated using the mean signal values of methylation probes mapping to the same gene. In other words, β -values resemble the mean methylation level of individual genes in a given sample. Next, for each gene, we correlated its TACNA expression level with its mean methylation level across all samples. We observed that for a subset of genes, their TACNA expression levels correlate with their mean methylation levels (r range = -0.65 to 0.45, see panel b of figure below). These results indicate that DNA methylation could be one of the underlying mechanisms driving the degree of transcriptional adaptation to CNAs.

To highlight these results, we included the following supplementary figure to our manuscript:

Supplementary Fig. 9 a Manhattan plot showing the $-\log_{10}(P)$ enrichment value on the y-axis for the degree of transcriptional adaptation of genes per cytogenetic band defined according to the MSigDB Positional Gene Sets collection. The average degree of transcriptional adaptation was calculated for genes occurring once in a CNA-CES in both the GEO and TCGA dataset. The dotted lines represent a significance level of $P = 0.05$. **b** Spearman correlation between the mean methylation levels of individual genes and their degree of transcriptional adaptation in a subset of samples ($n = 9,317$) from the TCGA dataset.

The following text was added to the manuscript (page 11, lines 203-210):

“We observed that multiple genomic regions were enriched for genes having a higher or lower degree of transcriptional adaptation (Supplementary Fig. 9a and Supplementary Table 9). Epigenetic mechanisms such as DNA methylation are known to affect transcription levels of individual genes^{17,18}. When we explored available DNA methylation data for a subset of samples from the TCGA dataset ($n = 9,317$), we observed that for a subset of genes their TACNA expression levels correlated with their mean methylation levels (r range = $-0.65 - 0.45$, Supplementary Fig. 9b). These results indicate that DNA methylation could be one of the underlying mechanisms driving the degree of transcriptional adaptation to CNAs.”

We added the following text to the Supplementary Note (page 45):

“Association between DNA methylation and degree of transcriptional adaptation

For a subset of samples in the TCGA dataset ($n = 9,317$) available preprocessed methylation data generated with the Illumina 450 K array was collected. For each sample, we obtained the β -values of all individual genes. These β -values for individual genes were calculated using the mean signal values of methylation probes mapping to the same gene. In other words, β -values resemble the mean methylation level of individual genes in a given sample. For each gene, we correlated its mean methylation levels with TACNA expression levels across all samples.”

Remark 7: “How do adaptation profiles change with tissue/cancer type?”

We observed clear differences between TACNA profiles of different tumor types, which can be partially appreciated in the heatmap of the TACNA profiles of the TCGA dataset (Fig. 5c). For example, many renal clear cell carcinomas showed a pattern in their TACNA profile consistent with a deletion in chromosome 3p and an amplification in 5q, which have been reported as common observed genomic alterations in this tumor type.⁸ Likewise, common patterns consistent with a chromosome 10 deletion were observed in TACNA profiles of glioblastoma multiforme tumors, and a 1p/19q deletion in low-grade gliomas (see panels a to c of figure below).^{9,10} To facilitate the exploration of tissue specific TACNA profiles we generated heatmaps for every tumor type in combination with the tumor type specific average degree of TACNA plots and added them as Supplementary Fig. 12-20.

To explore the differences in the TACNA profile at the individual gene level we created the website www.genomicinstability.org (works best on Google Chrome). We apologize if you were not able to access this website due to a small error in our previously provided url. In the 'TACNA gene distribution' page a gene can be searched and subsequent the distribution of TACNA expression values over all tumor types is visualized. As an example, we provide a screenshot of the result of searching for ERBB2 in the TCGA dataset (see panel d of figure below).

(While addressing this remark we noticed that figure 5c contained an error, we therefore corrected this figure in the manuscript).

We added the following supplementary figure to our manuscript:

Supplementary Fig. 11 **a** TACNA profiles for renal clear cell carcinoma in the TCGA dataset. **b** TACNA profiles for glioblastoma multiforme in the TCGA dataset. **c** TACNA profiles for low-grade glioma in the TCGA dataset. **d** Example of exploring *ERBB2* TACNA values across samples in the TCGA dataset at www.genomicinstability.org. The bars represent the interquartile range.

The following text was added to the manuscript (page 13, lines 249-256):

“While exploring the landscape of transcriptional effects of CNAs, we observed common patterns consistent with well-known genomic alterations. For instance, many renal clear cell carcinomas in the TCGA dataset contained a pattern consistent with a deletion in chromosome 3p and an irregular amplification in chromosome 5q (Supplementary Fig. 11a)²⁴. Likewise, we observed patterns consistent with a chromosome 10 deletion in glioblastoma multiforme, and a 1p/19q deletion in low-grade gliomas (Supplementary Figs. 11b and 11c)^{25,26}. The landscape of transcriptional effects of CNAs can be explored at www.genomicinstability.org (Supplementary Fig. 11d).”

Remark 8: “The authors propose that a proportion of CNAs colocalized within common fragile sites, yet the location of most common fragile sites have only been mapped using low resolution methods. This should be further addressed or discussed.”

According to the remark made by the referee, in our results section we now state that the fragile sites were mapped using low resolution methods (page 8, lines 156-159):

“Although the location of most of these common fragile sites have been mapped using low-resolution methods, these results are compatible with the notion of genomic instability underlying structural abnormalities such as CNAs.”

Referee #2

Referee #2 made the following suggestions regarding our manuscript:

Remark 9: “The ICA and TACNAS approaches are described in detail in the supplementary note with an accompanying figure. However, it would be very helpful to give an exposition of the approach and a high-level overview at the start of the main text/results. Considering this represents the major advancement, along with its findings, it would set the tone for the rest of the article as well as place things in a more interpretable context.”

To provide a better overview of our approach, we now included Supplementary Fig. 1 as a panel in Fig. 1.

Fig. 1 c Identification of underlying regulatory factors of the mRNA transcriptome. We hypothesized that the observed gene expression in a gene expression profile is the result of (i) the effect of underlying regulatory factors (i.e. source signals) on expression levels of individual genes and (ii) the activity of these underlying regulatory factors in a complex biopsy (i.e. mixing matrix). ICA was used to capture the number and nature of these underlying

regulatory factors for all four datasets separately. This resulted in estimated sources, representing the effects of independent underlying regulatory factors on the expression levels of individual genes, and a mixing matrix reflecting the activity of each estimated source in each gene expression profile. ICA was run 25 times with random initialization, followed by consensus sources estimation using a credibility index $\geq 50\%$.

Additionally, we added a figure panel describing our approach to TACNA profiling in more detail to Fig. 2:

Fig 2. a Transcriptional adaptation to CNA (TACNA) profiling. For each dataset, weights of genes mapping to a contiguous genomic region in CNA-CESs marked by the detection algorithm were retained in the CES matrix. Weights of genes that mapped outside marked genomic regions were instead set to zero. Next, TACNA profiles were calculated as the product between this transformed CES matrix and the consensus mixing matrix.

We added the following text to the manuscript (page 6, lines 112-116):

“Weights of genes mapping to a contiguous genomic region in CNA-CESs marked by the detection algorithm were retained in the CNA-CES matrix. Weights of genes that mapped outside marked genomic regions in a CNA-CES were set to zero. Next, TACNA profiles were calculated as the product between this transformed CNA-CES matrix and the consensus mixing matrix (Fig. 2a).”

Remark 10: “The authors demonstrate that the identified CNAs predicted from the TACNAS approach are highly correlated on average with actual CNA measured within the same dataset. Could the authors also show that this finding is robust to a cross-validation approach?”

To address this remark, we conducted a five-fold cross-validation analysis using the TCGA dataset and CCLE dataset using the following steps, which we added as a new section to the Supplementary Note:

- Samples from the expression datasets were randomly divided into five groups using a multinomial distribution simulation. The total number of samples in each of the five groups was 2160, 2139, 2204, 2163, and 2151, respectively for the TCGA dataset. The total number of samples in each of the five groups was 217, 218, 217, 204, and 211, respectively for the CCLE dataset.

- Gene expression profiles from the mRNA expression dataset were standardized on the gene-level to a mean of zero and a standard deviation of one (*standardized_mRNA*).
- For the i^{th} fold the following steps were conducted ($i = 1,2,3,4$ and 5):
 - The input dataset ($mRNA_{CV_i}$) for next steps was obtained by excluding samples in the i^{th} group from the unstandardized mRNA expression dataset.
 - Gene expression profiles from $mRNA_{CV_i}$ were standardized to a mean of zero and standard deviation of one.
 - Consensus independent component analysis was applied on the standardized $mRNA_{CV_i}$ to obtain consensus estimated sources matrix ($CESM_i$).
 - $CESM_i$ and *standardized_mRNA* were used to obtain the consensus mixing matrix (CMM_i):

$$CMM_i = ((CESM_i)' \times CESM_i)^{-1} \times (CESM_i)' \times \textit{standardized_mRNA}$$
 - $CESM_i$ with extreme valued contiguous genomic regions (CNA_{CESM_i}) were identified using the DEGR algorithm as described in our original Supplementary Note section.
 - CNA_{CESM_i} and the CMM_i were used to obtain TACNA profiles for the samples present in $mRNA_{CV_i}$ along with the samples in the i^{th} group ($TACNAP_{excluded_i}$).
 - Pearson correlation coefficients were calculated between $TACNAP_{excluded_i}$ and corresponding CNA profiles (derived from SNP arrays).

The correlation coefficients between $TACNAP_{excluded_i}$ and their paired CNA profiles had a distribution mode $r > 0.54$ for TCGA in all four folds and distribution mode $r > 0.45$ for CCLE in all five folds of the cross-validation analysis (see figures below). Due to time constraint, we could manage to get results for 4 different folds for TCGA but for CCLE, we managed to conduct 5 folds of cross-validation analysis. These results show that TACNA-profiling is robust to cross-validation analysis within one platform. As patient-derived tumor gene expression profiles are known to be heterogeneous in CNA occurrence, exclusion of any number of samples is expected to create difference in CNA-consensus estimated sources (CNA-CESs). In other words, it can happen that the 20% samples left out in one fold of the cross-validation analysis contain a CNA that is not present in the 80% of the samples used to find the CNA-CESs. Therefore, when constructing the TACNA profiles for the 20% of the left-out samples, the effect of this specific CNA on gene expression levels would not be present. As a result, the correlations observed in the cross-validation analysis are slightly lower than the correlations we found between TACNA profiles and their paired CNA profiles in our previous analysis using all samples from the TCGA dataset.

Based on our cross-validation analysis, we added the following supplementary figures:

Supplementary Fig. 6 Cross-validation analysis in the TCGA dataset. Distributions of Pearson correlation coefficients between TACNA profiles of randomly chosen 20% of samples of the TCGA dataset (RNA sequencing) in each fold using CNA-CEs derived from the remaining 80% of samples of the TCGA dataset and paired CNA profiles (derived from SNP).

Supplementary Fig. 7 Cross-validation analysis in the CCLE dataset. Distributions of Pearson correlation coefficients between TACNA profiles of randomly chosen 20% of samples of the CCLE dataset (microarray) in each fold using CNA-CESs derived from the remaining 80% of samples of the CCLE dataset and paired CNA profiles (derived from SNP).

Based on our cross-validation analysis, we added the following text to the Supplementary Note (page 38):

“Cross-validation analysis within one platform

A cross-validation analysis of TACNA-profiling was conducted to test the robustness of this method within one platform. A five-fold cross-validation analysis an mRNA expression dataset was done using the following steps:

- Samples from the mRNA expression dataset were randomly divided into five groups using a multinomial distribution simulation.
- Gene expression profiles from the mRNA expression dataset were standardized on the gene-level to a mean of zero and a standard deviation of one (*standardized_mRNA*).
- For the i^{th} fold the following steps were conducted ($i = 1,2,3,4$ and 5):
 - The input dataset ($mRNA_{CV_i}$) for next steps was obtained by excluding samples in the i^{th} group from the unstandardized mRNA expression dataset.
 - Gene expression profiles from $mRNA_{CV_i}$ were standardized to a mean of zero and standard deviation of one.
 - Consensus independent component analysis was applied on the standardized $mRNA_{CV_i}$ to obtain consensus estimated sources matrix ($CESM_i$).
 - $CESM_i$ and *standardized_mRNA* were used to obtain the consensus mixing matrix (CMM_i):

$$CMM_i = ((CESM_i)' \times CESM_i)^{-1} \times (CESM_i)' \times \textit{standardized_mRNA}$$
 - $CESM_i$ with extreme valued contiguous genomic regions (CNA_{CESM_i}) were identified using the DEGR algorithm as described in the Supplementary Note.
 - CNA_{CESM_i} and the CMM_i were used to obtain TACNA profiles for the samples present in $mRNA_{CV_i}$ along with the samples in the i^{th} group ($TACNAP_{excluded_i}$).
 - Pearson correlation coefficients were calculated between $TACNAP_{excluded_i}$ and corresponding CNA profiles (derived from SNP arrays)."

Remark 11: "While the authors showed that -weights- are correlated for some genes across studies, it would be very convincing if the authors could also perform a cross-study cross-validation."

To address this remark, we conducted cross-study cross-validation analyses on the TCGA, CCLE, and GDSC mRNA expression datasets (${}^3C_2 = 6$ cross-study cross-validation analysis). These datasets were chosen as we also obtained paired CNA profiles (independently generated with SNP arrays) of samples from these datasets. Expression data of the CCLE and GDSC datasets were transformed from probe-level to gene-level using the 'jetset' package in R. Details about the 'jetset' package are provided in the Supplementary Note. Expression data of TCGA dataset was already provided on gene-level. The following steps were conducted (and described in new section in the Supplementary Note) for each cross-study cross-validation analysis where

consensus estimated sources of dataset i were used to obtain TACNA-profiles of dataset j ($i, j =$ TCGA, CCLE & GDSC):

- Genes not present in both dataset i and j were removed from the analysis.
- Both dataset i and dataset j were standardized on gene-level separately, which means each gene expression was transformed to a mean of zero and standard deviation of one.
- Both of these standardized datasets were sample-wise merged to obtain a combined dataset (*Combined_iused_for_j*).
- Consensus estimated sources matrix of dataset i (CEM_i) and *Combined_iused_for_j* were used to obtain consensus mixing matrix (*CMMcombined_iused_for_j*).
$$CMM_{combined_i_used_for_j} = ((CEM_i)' \times CEM_i)^{-1} \times (CEM_i)' \times Combined_i_used_for_j$$
- CEM_i with extreme valued contiguous genomic region (CNA_{CEM_i}) were identified using DEGR algorithm.
- CNA_{CEM_i} and *CMMcombined_iused_for_j* enabled us to obtain TACNA-profiles for the samples present in dataset i along with the samples in the dataset j (*TACNAP_jusing_i*).
- Pearson correlation coefficient between *TACNAP_jusing_i* and corresponding copy number profiles (derived from SNP arrays) were obtained.

The correlation coefficients between *TACNAP_jusing_i* and their paired CNA profiles had a distribution mode $r > 0.5$ in all 6 cross-study cross-validation analyses (see figure below). These results show that the CNA_{CEM} of a mRNA expression dataset from one platform (e.g. RNA-seq) can be used to project on the mRNA expression dataset from another platform (e.g. microarray) even when the platform uses different technology. Additionally, these results show that the CNA_{CEM} of a mRNA expression dataset from patient-derived samples can be used to project on the mRNA expression dataset from *in vitro* samples and vice versa. However, cross-dataset heterogeneity in CNA occurrence is still a constraint to reconstruct TACNA profiles of a dataset using CNA_{CEM} of any other dataset.

Based on our cross-study cross-validation, we added the following supplementary figure:

Supplementary Fig. 8 Cross-study cross-validation. Distributions of Pearson correlation coefficients between TACNA profiles of dataset *i* using CNA-CEs derived from dataset *j* and paired CNA profiles (derived from SNP) of dataset *i* (*i* by *j* where *i* and *j* are from TCGA, CCLE & GDSC).

Based on our cross-validation and cross-study cross-validation, the following text was added to the manuscript (page 9, lines 178-181):

“In addition, TACNA profiling was robust to both cross-validation within one platform (Supplementary Figs. 6 and 7, mode of distribution r range = 0.45-0.55) and cross-platform cross-validation (Supplementary Fig. 8, mode of distribution r range = 0.50-0.61, see Supplementary Note for details).”

Based on our cross-study cross-validation analyses, we added the following text to the Supplementary Note (page 39):

“Cross-study cross-validation analysis

A cross-study cross-validation analysis of TACNA-profiling was conducted to test the robustness of this method across different studies or platforms. Following steps were conducted for each cross-study cross-validation analysis where consensus estimated sources of dataset i were used to obtain TACNA-profiles of dataset j :

- Genes not present in both dataset i and j were removed from the analysis.
- Both dataset i and dataset j were standardized on gene-level separately, which means each gene expression is transformed to a mean of zero and standard deviation of one.
- Both of these standardized datasets were sample-wise merged to obtain a combined dataset (*Combined_iused_for_j*).
- Consensus estimated sources matrix of dataset i ($CESM_i$) and *Combined_iused_for_j* were used to obtain consensus mixing matrix ($CMM_{combined_i_used_for_j}$).
$$CMM_{combined_i_used_for_j} = ((CESM_i)' \times CESM_i)^{-1} \times (CESM_i)' \times Combined_i_used_for_j$$
- $CESM_i$ with extreme valued contiguous genomic region (CNA_CESM_i) were identified using DEGR algorithm.
- CNA_CESM_i and $CMM_{combined_i_used_for_j}$ were used to obtain TACNA-profiles for the samples present in dataset i along with the samples in the dataset j (*TACNAP_jusing_i*).
- Pearson correlation coefficient between *TACNAP_jusing_i* and corresponding copy number profiles (derived from SNP arrays) were obtained.

Cross-dataset heterogeneity in CNA occurrence is a constraint to reconstruct TACNA profiles of a dataset using CNA_CESM of any other dataset.”

Remark 12: “The authors take care to clean and prepare the data as much as possible, but it is not clear how various relevant covariates that should drive large heterogeneity into account.”

We appreciate the point raised by the referee. When performing consensus-ICA on the mRNA transcriptome, independent underlying regulatory factors of the transcriptome are identified as

CESs. These underlying regulatory factors include not only the downstream effect of CNAs, but also other relevant covariates that drive large heterogeneity in the mRNA transcriptome such as study-specific batch effects, gender etc. For TACNA profiling, we regenerate gene expression profiles by using only CESs (i.e. underlying regulatory factors) that were identified by our detection algorithm representing downstream effects of CNAs on mRNA levels. By including only these CESs, we automatically excluded other non-genetic covariates driving heterogeneity in the mRNA from our pipeline for TACNA profiling. For example, in the GEO dataset we identified a CES that likely drives heterogeneity in mRNA expression on the X and Y chromosomes because of differences in gender (see Supplementary Fig. 1). This demonstrates that consensus-ICA can effectively identify these types of covariates and their genomic effects. Incidentally, this particular CES influences a contiguous genomic region which makes it a candidate CNA-CES. However, for TACNA profiling we excluded the X and Y chromosomes which in turn negates the influence of this gender CES. Below, we show the mixing matrix weights (i.e. indirect activity measurements) of this underlying regulatory factor in individual samples, showing that it clearly discriminates between males and females.

Supplementary Fig. 1 Gender CES. A density plot is shown for the mixing matrix weights of CES 471 in the GEO dataset. Males ($n = 1,074$), females ($n = 1,300$) and unknown biological gender status (19,218) is plotted separately depending on the available information provided online by the original authors (GEO).

To emphasize how we dealt with relevant covariates in our analysis, we added the following text to our manuscript (page 6, lines 109-112):

“By including only these CNA-CESs, we automatically excluded other CESs that capture non-genetic factors that can affect gene expression levels from our pipeline for TACNA profiling (see Supplementary Fig. 1 and Supplementary Note)”.

We added the following text to the Supplementary Note (page 45):

“Identification of CES capturing gender differences

Biological gender annotation for a subset of samples from the GEO dataset was collected from the GEO portal. A Mann-Whitney U test was performed for every CES comparing the mixing matrix weights of samples annotated to be male and samples annotated to be female. CES 471 had the highest discrimination power between male and female samples (AUC = 0.99).”

Remark 13: “For genes identified using TACNA where correspond data have actual CNA or imputed CNA from genotype, can the authors perform a QTL analysis while controlling for relevant variables? There was some work showing that CNAs exist in more extreme regions, but it would be informative to see that actual CNA variation is statistically linked with expression values at these regions.”

In order to address this remark, we conducted expression quantitative trait loci (eQTL) analyses using the mRNA expression profiles, the TACNA profiles and the CNA profiles (derived from SNP arrays) from the TCGA dataset ($n = 10,817$). Details are given below:

- Genes and samples not present in all three of the above-mentioned profiles were removed from the analysis.
- Pearson correlation coefficients were obtained between mRNA expression profiles and CNA profiles on the gene-level.
- Pearson correlation coefficients were obtained between TACNA profiles and CNA profiles on the gene-level.
- Association within the CNA profiles on the gene-level was computed using Pearson correlation coefficient.
- Partial correlation coefficients between TACNA profiles and CNA profiles were computed on the gene-level to identify *trans* effect. Partial correlation analysis was conducted to remove false positive *trans* effects driven by CNA co-occurrence. For example, it could happen that with Pearson correlation analysis, we find *trans* effect between CNA_x on chromosome 1 and mRNA expression of $gene_z$ on chromosome 5. However, this effect could be explained by the fact that an underlying CNA_y on chromosome 5 (where $gene_z$ is mapping to) always co-occurs with the CNA_x on chromosome 1. The observed *trans* effect is then actually a *cis* effect between CNA_y and $gene_z$.

In general, we observed a stronger correlation between TACNA profiles and CNA profiles compared to the correlation between mRNA expression profiles and CNA profiles on the gene-level (see panel a and b of figure below). This indicates that TACNA profiling increases the statistical power to detect associations between CNAs and mRNA expression levels.

In addition, we observed a clear association between the degree of TACNA and the correlation between CNAs and regular mRNA levels (Spearman $r = -0.83$, see panel e of figure below). For genes with a high degree of transcriptional adaptation we observed low correlation. In contrast, for genes with a low degree of transcriptional adaptation we observed higher correlations between CNAs and regular mRNA levels.

Based on our eQTL analyses, we added the following supplementary figure:

Supplementary Fig. 4 **a** Correlation between mRNA expression profiles and CNA profiles (derived from SNP) from TCGA dataset on the gene-level. **b** Correlation between TACNA profiles and CNA profiles (derived from SNP) from TCGA dataset on the gene-level. **c** Association within CNA profiles (derived from SNP) from TCGA dataset on the gene-level. **d** Partial correlation coefficients between TACNA profiles and CNA profiles (derived from SNP) from TCGA dataset on the gene-level. Insets of each panel represents correlation coefficients for genes mapping to chromosome 1. **e** Scatter plot of the degree of TACNA versus Pearson correlations between mRNA and SNP expressions for genes occurring once in an extreme-valued region of a CNA-CES in the TCGA dataset. Only CNA-CESs with >50 genes in their extreme-valued region were considered.

Based on our eQTL analyses, the following text was added to the manuscript (page 8, lines 147-153):

“We conducted expression quantitative trait loci (eQTL) analyses using the mRNA expression profiles, the TACNA profiles and the CNA profiles (derived from SNP arrays) from the TCGA dataset ($n = 10,817$). In general, we observed a stronger correlation between TACNA profiles and CNA profiles compared to the correlation between mRNA expression profiles and CNA profiles on the gene-level (Supplementary Fig. 4). This indicates that TACNA profiling increases statistical power to detect eQTLs, i.e. associations between CNAs and mRNA expression levels.”

We added the following text to the Supplementary Note (page 46):

“Expression quantitative trait loci (eQTL) analyses

eQTL analysis was conducted using the mRNA expression profiles, the TACNA profiles and the CNA profiles of the TCGA dataset ($n = 10,817$). Details are given below:

- Genes and samples not present in all three of the above-mentioned datasets were removed from the analysis.
- Pearson correlation coefficients were obtained between mRNA expression profiles and CNA profiles on the gene-level.
- Pearson correlation coefficients were obtained between TACNA profiles and CNA profiles on the gene-level.
- Association within the CNA profiles on the gene-level was computed using Pearson correlation coefficient.
- Partial correlation coefficients between TACNA profiles and CNA profiles were computed on the gene-level to identify *trans* effect. Partial correlation analysis was conducted to remove false positive *trans* effects driven by CNA co-occurrence.”

References

1. Stingele, S. *et al.* Global analysis of genome, transcriptome and proteome reveals the

- response to aneuploidy in human cells. *Mol. Syst. Biol.* **8**, 608 (2012).
2. Fehrmann, R. S. N. *et al.* Gene expression analysis identifies global gene dosage sensitivity in cancer. *Nat. Genet.* **47**, 115–125 (2015).
 3. Birchler, J. A. & Veitia, R. A. Gene balance hypothesis: connecting issues of dosage sensitivity across biological disciplines. *Proc. Natl. Acad. Sci. U. S. A.* **109**, 14746–14753 (2012).
 4. Giurgiu, M. *et al.* CORUM: The comprehensive resource of mammalian protein complexes - 2019. *Nucleic Acids Res.* **47**, D559–D563 (2019).
 5. Subramanian, A. *et al.* Gene set enrichment analysis: a knowledge-based approach for interpreting genome-wide expression profiles. *Proc. Natl. Acad. Sci. U. S. A.* **102**, 15545–15550 (2005).
 6. Baylin, S. B. DNA methylation and gene silencing in cancer. *Nat. Clin. Pract. Oncol.* **2**, S4–S11 (2005).
 7. Du, J., Johnson, L. M., Jacobsen, S. E. & Patel, D. J. DNA methylation pathways and their crosstalk with histone methylation. *Nat. Rev. Mol. Cell Biol.* **16**, 519–532 (2015).
 8. Mitchell, T. J. *et al.* Timing the landmark events in the evolution of clear cell renal cell cancer: TRACERx Renal. *Cell* **173**, 611–623.e17 (2018).
 9. Fults, D. & Pedone, C. Deletion mapping of the long arm of chromosome 10 in glioblastoma multiforme. *Genes, Chromosom. Cancer* **7**, 173–177 (1993).
 10. Eckel-Passow, J. E. *et al.* Glioma groups based on 1p/19q, IDH, and TERT promoter mutations in tumors. *N. Engl. J. Med.* **372**, 2499–2508 (2015).

REVIEWERS' COMMENTS:

Reviewer #1 (Remarks to the Author):

I commend the authors for addressing my concerns and questions in a very thorough and thoughtful manner. I have no additional concerns and highly recommend this manuscript for publication in Nature Communications.

Reviewer #2 (Remarks to the Author):

The authors have done a considerable amount of work to address all comments and I thank them for their effort. I have no further comments.